



# Evaluating spectral cloud effective radius retrievals from the Enhanced MODIS Airborne Simulator (eMAS) during ORACLES

Kerry Meyer[1], Steven Platnick[1], G. Thomas Arnold[2,1], Nandana Amarasinghe[2,1], Daniel Miller[3,1], Jennifer Small-Griswold[4], Mikael Witte[5], Brian Cairns[6], Siddhant Gupta[7], Greg McFarquhar[8], Joseph O'Brien[7]

[1]NASA Goddard Space Flight Center, Greenbelt, 20771, USA
[2]SSAI, Lanham, 20706, USA
[3]GESTAR-II, University of Maryland-Baltimore County, Baltimore, 21250, USA
[4]Department of Atmospheric Sciences, University of Hawaii at Manoa, Honolulu, 96822, USA
[5]Department of Meteorology, Naval Postgraduate School, Monterey, 93943, USA
[6]NASA Goddard Institute for Space Studies, New York, 10025, USA
[7]Argonne National Laboratory, Lemont, 60439, USA
[8]School of Meteorology, University of Oklahoma, Norman, 73072, USA

*Correspondence to*: Kerry Meyer (kerry.meyer@nasa.gov)

**Abstract.** Satellite remote sensing retrievals of cloud effective radius (CER) are widely used for studies of aerosol/cloud interactions. Such retrievals, however, rely on forward radiative transfer (RT) calculations using simplified assumptions that can lead to retrieval errors when the real atmosphere deviates from the forward model. Here, coincident airborne remote sensing and in situ observations obtained during NASA's ObseRvations of Aerosols above CLouds and their intEractionS (ORACLES) field campaign are used to evaluate retrievals of CER for marine boundary layer stratocumulus clouds and to explore impacts of forward RT model assumptions and other confounding factors. Specifically, spectral CER retrievals from the Enhanced MODIS Airborne Simulator (eMAS) and the Research Scanning Polarimeter (RSP) are compared with polarimetric retrievals from RSP and with CER derived from droplet size distributions (DSDs) observed by the Phase Doppler Interferometer (PDI) and a combination of the Cloud and Aerosol Spectrometer (CAS) and two dimensional Stereo probe (2D-S). The sensitivities of the eMAS and RSP spectral retrievals to assumptions on the DSD effective variance (CEV) and liquid water complex index of refraction are explored. CER and CEV inferred from eMAS spectral reflectance observations of the backscatter glory provide additional context for the spectral CER retrievals. The spectral and polarimetric CER retrieval agreement is case dependent, and updating the retrieval RT assumptions, including using RSP polarimetric CEV retrievals as a constraint, yields mixed results that are tied to differing sensitivities to vertical heterogeneity. Moreover, the in situ cloud probes, often used as the benchmark for remote sensing CER retrieval assessments, themselves do not agree, with PDI DSDs yielding CER 1.3-1.6 µm larger than CAS and CEV roughly 50-60% smaller than CAS. Implications for the interpretation of spectral and polarimetric CER retrievals and their agreement are discussed.





## 1 Introduction

The Earth's climate is strongly influenced by clouds. Cloud interactions with incoming shortwave solar radiation and emitted longwave terrestrial radiation, reflecting the former (cooling effect) and trapping the latter (warming effect), are the largest modulator of the Earth's radiative budget. These radiative effects themselves can be altered by cloud microphysical interactions

with atmospheric aerosols that can change cloud albedo (Costantino and Breon, 2013; Costantino and Bréon, 2010; Gupta et al., 2021; Twomey, 1974, 1977), cloud lifetime (Albrecht, 1989), and precipitation (Chen et al., 2011; Gupta et al., 2022a; van den Heever et al., 2006; van den Heever and Cotton, 2007; Martins et al., 2009), having direct implications on the Earth's hydrological cycle. While much is known about clouds, a complete understanding of the role of clouds in the Earth's climate and on cloud feedbacks in response to climate change on regional and global scales remains a challenge that is the leading

contributor to inter-model uncertainty in climate sensitivity (Dufresne and Bony, 2008; Webb et al., 2012; Zelinka et al., 2012, 2017).

Satellite remote sensing retrievals of cloud properties enable the characterization of clouds, and studies of cloud processes, over large spatial and temporal scales. Of particular note are cloud retrievals using spectral imagery from polar-orbiting

satellites that provide global observations such as the Advanced Very High Resolution Radiometer (AVHRR) on NOAA's heritage operational weather satellites (e.g., Heidinger et al., 2014), the Moderate-resolution Imaging Spectroradiometer (MODIS) on NASA's Terra and Aqua satellites (Minnis et al., 2021; Platnick et al., 2017), and the Visible Infrared Imaging Radiometer Suite (VIIRS) on the NASA/NOAA Suomi NPP satellite and the new generation of NOAA weather satellites (e.g., Frey et al., 2020; Platnick et al., 2021). These sensors observe reflected shortwave and emitted longwave radiation in

narrowband spectral channels that provide rich information on cloud detection, cloud-top properties (Baum et al., 2012; Menzel et al., 2008; Wylie and Menzel, 1999), and cloud optical/microphysical properties that include cloud optical thickness (COT) and cloud effective particle radius (CER) (e.g., Twomey and Cocks, 1982). CER in particular has been widely used for studies of the impacts of aerosol/cloud interactions on radiation (Oreopoulos and Platnick, 2008; Platnick and Oreopoulos, 2008), liquid cloud droplet concentration (Grosvenor et al., 2018), and precipitation (Braga et al., 2021; Rosenfeld et al., 2012).


Imager remote sensing retrievals of CER, from satellites or airborne platforms, typically occur simultaneously with retrievals of COT using a bi-spectral approach pairing reflectance in a non-absorbing visible (VIS), near-infrared (NIR), or shortwave infrared (SWIR) spectral channel sensitive to COT with reflectance in an absorbing SWIR or mid-wave IR (MWIR) channel sensitive to CER (Nakajima and King, 1990). In practice, the observed reflectance-to-COT/CER retrieval inversion relies on

forward radiative transfer (RT) calculations using simplified assumptions, for instance a single, plane-parallel cloud layer having a horizontally and vertically homogeneous particle size distribution. These simplifications, while pragmatic for global-scale retrievals where computational efficiency is a key requirement, nevertheless can lead to retrieval errors for both liquid clouds (Marshak et al., 2006; Várnai and Marshak, 2002; Zuidema and Evans, 1998), whose droplet sizes typically increase



with height due to adiabatic growth, and ice clouds (Wang et al., 2019; Zhang et al., 2010), whose crystal sizes typically
decrease with height due to sedimentation of larger particles, or for scenes having broken or otherwise spatially heterogeneous
cloud fields (Zhang et al., 2012, 2016). Moreover, for liquid clouds, recent studies have shown that the imaginary part of the
complex index of refraction of liquid water, a fundamental assumption for RT calculations, has a temperature dependence in
the SWIR (Kou et al., 1993) that has sizeable impacts on computed cloudy reflectance in this spectral region and thus on
spectral retrievals of CER (Platnick et al., 2020).


Given their utility and wide use, numerous efforts have been undertaken to evaluate satellite cloud remote sensing retrievals.
For retrievals of liquid CER, defined as the ratio of the third to second moments of the particle size distribution, evaluation
efforts for liquid clouds often include comparisons against cloud droplet size distributions (DSDs) measured in situ by airborne
cloud probes. Such evaluations against cloud probes have a long history (Gupta et al., 2022b; King et al., 2013; Min et al.,
2012; Nakajima et al., 1991; Noble and Hudson, 2015; Painemal and Zuidema, 2011; Platnick and Valero, 1995) and
consistently find that bi-spectral CER retrievals, while highly correlated with CER derived from in situ DSDs, nevertheless
are on average roughly 2μm larger than the in situ measurements. In contrast, (Witte et al., 2018), using DSDs measured by
the Phase Doppler Interferometer (PDI) (Chuang et al., 2008) during three different field campaigns, found no systematic bias
in MODIS CER retrievals, which could be interpreted as the CER derived from legacy probe DSDs being biased low due to
inadequate characterization of the full width of the DSD.

The ObseRvations of Aerosols above CLouds and their intEractionS (ORACLES) field campaign (Redemann et al., 2021), a
NASA Earth Venture Suborbital-2 investigation with three deployments from 2016 to 2018, provides another opportunity to
evaluate imager bi-spectral CER retrievals against cloud probe DSDs. ORACLES targeted the unique aerosol and cloud
environment over the southeast (SE) Atlantic Ocean where an extensive biomass burning smoke layer overlies a quasi-
permanent marine stratocumulus cloud deck (Alfaro-Contreras et al., 2014a; Devasthale and Thomas, 2011; Meyer et al., 2013;
Swap et al., 2003). The 2016 ORACLES deployment featured two aircraft: the low-altitude P-3 Orion hosting a comprehensive
suite of aerosol, cloud, and atmospheric chemistry and meteorological in situ instrumentation, and the high-altitude ER-2
hosting a diverse active and passive remote sensing payload. The P-3 in situ instruments included the PDI cloud probe, the
Cloud and Aerosol Spectrometer (CAS) (Baumgardner et al., 2001) cloud probe on the Cloud, Aerosol, and Precipitation
Spectrometer (CAPS), the Cloud Droplet Probe (CDP) (Lance et al., 2010), the Two-Dimensional Stereo Probe (2D-S)
(Lawson et al., 2006), the High Volume Precipitation Sampler (HVPS-3) (Lawson et al., 1998), and the King hot wire (King
et al., 1978) to measure the bulk liquid water content. The ER-2 remote sensing payload included multi-spectral imagery from
the Enhanced MODIS Airborne Simulator (eMAS) (Ellis et al., 2011; King et al., 1996) and multi-angle polarimetry from the
Research Scanning Polarimeter (RSP) (Cairns et al., 1999, 2003). eMAS has SW and LW spectral channels analogous to
MODIS, VIIRS, and other satellite imagers, enabling bi-spectral COT/CER retrievals using channel pairs featuring several
SWIR channels in both the 1.6 and 2μm spectral regions. RSP's along-track hyper-angle polarimetric observations provide



independent retrievals of CER along with the effective variance (CEV) of the DSD for observations that sample the scattering angles around the cloud-bow, and its spectral channel complement also enables multi-angle bi-spectral COT/CER retrievals. During ORACLES 2016, multiple science flights featured coordinated maneuvers where the P-3 sampled within the stratocumulus cloud layer while the ER-2 made multiple passes overhead. These coordinated flights enable an evaluation of remote sensing retrievals, and key retrieval assumptions, against spatially and temporally co-located cloud probe measurements using distinctly different in situ sampling techniques.

In this paper, we provide a brief overview of the multi-spectral imager eMAS, its operations during ORACLES 2016, including spectral channel configuration changes and their impacts on eMAS science products, and post-deployment radiometric calibration efforts. We then show results of an extensive evaluation of bi-spectral CER retrievals from eMAS and RSP against polarimetric CER retrievals from RSP and CER derived from the in situ cloud probes. In addition, in several instances during the campaign the ER-2 flight direction was oriented such that eMAS observed scattering angles near the direct backscatter region within its swath, enabling the inference of CER and DSD effective variance from the total reflectance angular features of the "glory" akin to the polarized cloud-bow retrievals of RSP. We show results from these glory retrievals and conclude with a discussion of the broader implications of the consistency, or lack thereof, of in situ cloud probe observations and remote sensing retrievals of CER from spectral reflectance and polarimetry having different, though complementary, sensitivities and information content.

## 2 Data and Methodology

### 2.1 ER-2 Remote Sensing Observations

Two instruments in the ER-2 remote sensing payload are used here – the Enhanced MODIS Airborne Simulator (eMAS) and the Research Scanning Polarimeter (RSP). eMAS, a NASA facility instrument managed by the Ames Research Center (ARC) Airborne Sensor Facility (ASF), is a scanning spectrometer that measures reflected solar and emitted terrestrial radiation in 38 narrowband spectral channels between 0.47 and 14.1 μm wavelengths (Ellis et al., 2011; King et al., 1996). It was originally designed to provide an airborne platform for developing, testing, and refining geophysical retrieval algorithms for the Moderate-resolution Imaging Spectroradiometer (MODIS) prior to its launch on NASA's Terra and Aqua satellites. From a notional ER-2 flight altitude of 20 km, eMAS views a 37.25 km wide swath with a ground-level pixel size at nadir of roughly 50 m. In addition to broad swath imagery that provides scene context for nadir viewing ER-2 remote sensing instruments (e.g., lidars, radars, etc.) as well as co-located P-3 in situ instruments during targeted coordination, eMAS can also provide Level-2 geophysical retrievals of land and ocean surface and atmospheric parameters that for ORACLES include cloud and aerosol optical and microphysical properties (King et al., 2004, 2010; Meyer et al., 2016).



RSP, a Principal Investigator (PI) instrument from NASA's Goddard Institute for Space Studies (GISS), is a scanning
polarimeter that simultaneously observes both the total reflectance and its linear polarization state in nine narrowband spectral
channels from 0.41 to 2.26 μm (Cairns et al., 2003). Rather than imaging across track, RSP only scans in the along-flight track
direction, achieving a high angular resolution at the expense of an across-track swath, which is only a single pixel wide. With
a 14 mrad instantaneous field of view (FOV), RSP has a native ground-level pixel size at nadir of roughly 300 m from a
notional ER-2 flight altitude of 20 km, though the effective pixel size may be larger due to co-location of the multi-angle
observations to different target heights for cloud retrieval products.

**2.2 P-3 In Situ Observations**

We make use of various in situ probes in this study that were flown on the P-3 during ORACLES. Several cloud probes provide
independent observations of cloud droplet size distributions (DSDs) that are used to calculate CER for comparisons with the
eMAS and RSP retrievals. These cloud probes include the legacy Cloud and Aerosol Spectrometer (CAS) (Baumgardner et
al., 2001), the Two-Dimensional Stereo Probe (2D-S) (Lawson et al., 2006), and the Phase Doppler Interferometer (PDI)
(Chuang et al., 2008). Note that CAS and 2D-S, having sensitivities to different droplet size ranges (droplets up to 50 μm
diameter for CAS and roughly between 25-150 μm diameter for 2D-S), are merged to create a single microphysics DSD; PDI,
on the other hand, has sensitivity to droplets between 3-500 μm. The Cloud Droplet Probe (CDP) (Lance et al., 2010) was also
flown in ORACLES, though its 2016 data are affected by an optical misalignment issue (Gupta et al., 2022a) and are not used
here. A King hot wire (King et al., 1978) provides liquid water content (LWC) measurements that are used as constraints on
the CAS DSDs. All in situ data used in this study are reported at 1-second sampling and are obtained from the multi-instrument
merged product files pre-packaged by the ORACLES Science Team for ease of use: King LWC and PDI DSDs from a P-3
merged dataset product (Shinozuka, n.d.) and CAS/2D-S DSDs from a merged microphysics product (O'Brien et al., n.d.).

**2.3 Remote Sensing Cloud Property Datasets**

The standard eMAS and RSP cloud product datasets, produced by their respective science teams at NASA's Goddard Space
Flight Center (GSFC) and at GISS, are used here. For eMAS, the primary science products produced during and after field
campaign deployments are the geolocated and calibrated Level-1B (L1B) spectral radiances and the Level-2 (L2) cloud
geophysical retrieval products that include cloud masking and cloud-top and optical/microphysical properties. The L2 cloud
products have heritage with the MODIS Science Team cloud products for MODIS and the Visible Infrared Imaging Radiometer
Suite (VIIRS). Specifically, the cloud mask has heritage with the NASA MODIS Collection 6 cloud mask product (MOD35)
(Ackerman et al., 2008; Frey et al., 2008), the cloud-top property retrievals have heritage with the NOAA Algorithm Working
Group (AWG) PATMOS-x algorithm (based on CLAVR-x (Heidinger and Pavolonis, 2009)) that is part of the MODIS/VIIRS
CLDPROP continuity cloud products (Platnick et al., 2021), and the cloud optical/microphysical properties (e.g.,
thermodynamic phase, optical thickness, particle effective size, water path) have heritage with the NASA MODIS Collection
6 cloud product (MOD06) (Platnick et al., 2017). These L2 algorithms are part of the CHIMAERA shared-core suite of cloud



algorithms (Wind et al., 2020) that also includes the MODIS MOD06 and MODIS/VIIRS CLDPROP cloud optical property algorithms.

The along-track RSP data products produced by GISS include geolocated and calibrated L1B stokes-vector multi-angle
reflectances (I,Q,U) in nine spectral channels and L2 cloud geophysical retrieval products that feature cloud-top and optical/microphysical properties. As an additional step prior to the L2 cloud retrievals, there is an L1C product that reprojects and co-locates the L1B multi-angle reflectances to cloud-top height. This additional step is required to account for view angle parallax that causes targets at different altitudes in the atmosphere to shift relative to observations co-located to the surface. As a consequence, the L1C product requires prior cloud masking and cloud-top height retrievals, which are produced as an
interim geophysical retrieval obtained from the stereoscopic approach described in (Sinclair et al., 2017). The L2 cloud microphysical properties include retrievals of CER and CEV from the polarized cloud-bow (Alexandrov et al., 2012) and nadir only COT/CER retrievals from bi-spectral reflectance observations.

In addition to the above standard RSP cloud products produced by GISS, the GSFC CHIMAERA shared-core cloud algorithm
suite has been updated to provide bi-spectral COT/CER retrievals from RSP at each RSP observational angle. These multi-angle optical property retrievals are internally consistent with the standard eMAS retrievals in that they use the same science code, radiative transfer code and assumptions, and ancillary data. This consistency of approach enables synergistic use of the eMAS and RSP retrievals in the evaluations and science analyses shown here.

## 2.4 Computing Remote Sensing Cloud Effective Radius from In Situ Probe Observations

Comparing remote sensing retrievals of CER with those derived from in situ cloud probe measurements of droplet size distributions (DSDs) is not a straightforward enterprise. For instance, the spectral channels used for retrievals have differences in liquid water absorption that can yield quite different vertical penetration depths for the light scattered within the cloud. These spectral differences can lead to sensitivity to the vertical variation in the droplet size distributions and thus sensitivity to vertical variations in CER (e.g., Platnick, 2000). Furthermore, these vertically varying DSDs and associated spectral
sensitivities can be independently coupled to retrieval sensitivities to horizontal heterogeneity. As such, care must be taken to ensure that the in situ cloud probe sufficiently sampled the cloud layer and that the CER calculated from the probe observations reflects an appropriate estimate of what the imager spectral channel should ideally expect to retrieve from this known cloud profile. Thus, for the in situ cloud probe data, calculating the CER that would appropriately approximate an eMAS spectral CER retrieval is a multi-step process.


CER is defined as the ratio of the third and second moments of the DSD (Hansen and Travis, 1974). From cloud probe data, this is calculated as (Painemal and Zuidema, 2011),



$$CER = \frac{\sum_i r_i^3 \cdot n_i}{\sum_i r_i^2 \cdot n_i}, \tag{1}$$


where $n_i$ is the droplet number concentration (cm⁻³) per particle size bin $i$, and $r_i$ is the mean radius within the bin; $n_i$ and $r_i$ are obtained directly from the PDI or CAS/2D-S datasets, aggregated over 1 s intervals, in the ORACLES P-3 merged and microphysics files. For qualitative purposes, we compute CER for each 1 s probe DSD sample. Of more relevance for the CER retrieval evaluation, however, for each case study we also aggregate the 1 s PDI and CAS/2D-S DSD samples into single

vertical profiles having 10 m thick layers, from which we then compute a profile of CER for each probe at consistent vertical levels. This is accomplished first by calculating the layer DSD by averaging all 1 s sampled DSDs within the 10 m layer having corresponding probe LWC > 0.01 g/m³. Then the layer CER(z) is calculated from the layer DSD using Eqn. 1 above. This CER vertical profile is used as input to a forward radiative transfer (RT) model to calculate spectral vertical weighting functions from which estimates of the remote sensed eMAS spectral CER retrievals can be computed following (Platnick, 2000).


For the vertical weighting function computation, we use the 1-D, plane-parallel DISORT (Stamnes et al., 1988) as our RT model. Mie calculations using MIEV0 (Wiscombe, 1980) are used to calculate the spectrally resolved monodisperse single scattering properties of each PDI and CAS/2D-S size bin at high spectral resolution. The mean single scattering properties for each layer are subsequently computed by first weighting over the size dimension of the PDI or CAS/2D-S DSDs (i.e., $n_i$) and

then weighting spectrally with respect to the eMAS spectral response function ($f(\lambda)$) and spectral solar irradiance ($F_0(\lambda)$). The monochromatic bulk-extinction coefficient $\beta_{ext}(z,\lambda)$ for each layer, z, and wavelength, $\lambda$, is calculated as

$$\beta_{ext}(z,\lambda) = \sum_i \pi \cdot Q_e(z,\lambda,i) \cdot r_i^2 \cdot n_i(z), \tag{2}$$

where $Q_e(z,\lambda,i)$ is the extinction efficiency for the given wavelength, DSD bin center radius, and layer, and $n_i(z)$ is the droplet number concentration (cm⁻³) for the layer and DSD bin. The band-averaged extinction coefficient for eMAS spectral channel $b$ is then

$$\beta_{ext}(z,b) = \frac{\sum_\lambda \beta_{ext}(z,\lambda) \cdot f(\lambda) \cdot F_0(\lambda)}{\sum_\lambda f(\lambda) \cdot F_0(\lambda)} \tag{3}$$


Layer spectral COT for eMAS channel $b$, the vertical integral of the extinction coefficient over the layer, is then estimated as

$$COT(z,b) \cong \beta_{ext}(z,b) \cdot \Delta z, \tag{4}$$



where $\Delta z$ is the layer physical thickness (here, 10 m). We then input $COT(z,b)$ and the layer scattering properties into DISORT, along with the eMAS sun/satellite viewing geometry (solar and sensor zenith angles, relative azimuth), to iteratively calculate cumulative top-of-cloud reflectance stepping vertically through the cloud layer-by-layer starting at cloud top.

The radiative weighting functions are defined following (Platnick, 2000):

$$w_b(\tau) = \frac{1}{R(\tau_c)} \cdot \frac{dR(\tau)}{d\tau} \cong w_b(z) = \frac{1}{R(\tau_c)} \cdot \frac{R(z+\Delta z)-R(z)}{\tau(z+\Delta z)-\tau(z)}, \tag{5}$$

where $z$ here is physical depth into the cloud and is defined as zero at cloud top, $\Delta z$ is the physical thickness of the vertical layers (10 m), $\tau$ is the spectral COT at a given depth, $\tau_c$ is the total spectral COT of the entire sampled cloud depth, and $R$ is top-of-cloud reflectance. Thus, the weighting function essentially represents the incremental contribution of each layer to the 
cumulative cloud-top reflectance with respect to its incremental contribution to the increase in cloud optical thickness, normalized by the reflectance of the entire cloud, with calculation starting at cloud top and moving down. The estimate of the expected eMAS CER from spectral channel $b$ (e.g., 1.6μm, 2.13μm, 3.7μm, etc.) is then

$$CER_b = \sum_{z=z_t}^{z_b} CER(z) \cdot w_b(z) \cdot \left(\tau(z+\Delta z)-\tau(z)\right). \tag{6}$$


CER from Eqn. 6 can be thought of as a radiatively weighted average of the CER vertical profile observed by the cloud probes. (Platnick, 2000) showed that, for spectral channels in the 1.6, 2, and 3.7μm regions observed by imagers such as eMAS, MODIS, and VIIRS, the weighting functions in Eqn. 5 for liquid clouds peak near the top of the cloud (see also Figs. 10 (c) and 14 (c) below). Where these peaks are located vertically in relation to cloud top is determined by the spectral absorption 
strength (extinction coefficient in Eqns. 2-3), with more strongly absorbing spectral channels weighted closer to cloud top and more weakly absorbing channels weighted deeper within the cloud. While we attempt to explicitly account for varying spectral vertical sensitivities using Eqns. 2-6, previous studies often consider only a broad sensitivity to near-cloud top CER irrespective of spectral channel sensitivities. For instance, the (Gupta et al., 2022b) evaluation of MODIS CER against CAS and CDP DSDs observed in ORACLES averages the probe data over the topmost 10% of the geometrical thickness of the cloud.


In addition to evaluating CER retrievals, we also are interested in evaluating assumptions on the DSD that is used in the computation of cloud single-scattering properties that serve as inputs to the forward RT model and LUTs used in the retrieval inversion. For imager bi-spectral retrievals, the DSD typically is assumed to be a modified gamma distribution defined by a pair of scale and shape parameters, namely the effective radius (CER), i.e., the mean radius of the cross-section distribution, 
and the effective variance (CEV), i.e., the narrowness/symmetry of the distribution (small CEVs correspond to more narrow and less asymmetric distributions, and vice versa).



Following Eqn. 3 in (Miller et al., 2018), CEV can be estimated from in situ probe data as

$$CEV \equiv \frac{\langle r^4 \rangle \langle r^2 \rangle}{\langle r^3 \rangle^2} - 1, \tag{7}$$


where $\langle r^m \rangle = \sum_i r_i^m \cdot n_i$ is the $m$th moment of the DSD.

In this study, we show 1 s and aggregated 10 m layer CER and CEV calculated from both the CAS/2D-S and PDI probes. We note that (O'Brien et al., n.d.) selects CAS/2D-S to produce the best estimate merged microphysics product for ORACLES 2016 since they found that LWC computed from PDI overestimates the bulk and computed adiabatic LWC when considering all cloud samples. However, for samples near cloud top where the imager spectral CER retrievals are most sensitive, the PDI LWC matches more closely, and in fact slightly underestimates, the adiabatic LWC (see Fig. 3, (O'Brien et al., n.d.)), whereas CAS/2D-S significantly underestimates LWC, implying an underestimation of droplet size. (Gupta et al., 2022b) provides an adjustment to the magnitude of the CAS DSD size bins using the King LWC as a constraint that results in larger CER. However, these adjustments were derived via comparisons of bulk CAS and King LWC statistics across all in-cloud observations, regardless of position with respect to cloud top, obtained over multiple flights and still appear to underestimate droplet size near cloud top (see Figs. 10, 14 below).

## 3 eMAS Operations During ORACLES-2016

### 3.1 Deployment Overview

Flight operations during ORACLES-2016 were based out of Walvis Bay, Namibia, on the Atlantic coast, from late August through September. From Walvis Bay, the P-3 flew 13 science flights, beginning 30 August and ending 25 September, while the ER-2 flew 10 science flights, beginning 10 September and ending 27 September. Both aircraft operated solely over the SE Atlantic, in a region off the coasts of Angola and northern Namibia. Flight paths largely focused on targets of opportunity in addition to a frequent routine flight path that followed a diagonal from [20°S,10°E] to [10°S,0°E]. For the routine flight, the P-3 flew out and back along the diagonal, with frequent maneuvers to sample the environment above, below, and within both the aerosol and cloud layers; the ER-2, due to its longer range, typically flew the diagonal only on its return to Walvis Bay. Coordination between the two aircraft, i.e., direct ER-2 overflights of the P-3 for co-located in situ and remote sensing observations, was periodically pursued. In particular, two such coordinated flights, on 14 and 20 of September, featured multiple ER-2 overpasses while the P-3 was sampling within the cloud layer at two different locations on each day. Results from the 20 September co-locations are shown in Sect. 4.2.





The flight tracks of the 10 ER-2 science flights are shown in Fig. 1. Note that eMAS was not powered during the 22 September ER-2 flight (red track) due to a pre-flight data system issue. During the entirety of the ORACLES-2016 deployment, eMAS

operated without its full complement of spectral channels. Prior to pre-campaign integration on the ER-2, the 7.3 µm longwave channel became inoperable due to faulty wiring; thus, eMAS entered the field with only 37 spectral channels. Furthermore, issues with the cooling system for the cold optical bench caused periodic losses of the mid-wave/longwave (MW/LW) spectrometer and thus losses of thermal infrared (IR) channels. The first instance occurred prior to the 16 September flight, and only the shortwave spectrometer was powered for the flight. Post-flight hardware replacements allowed the MW/LW

spectrometer to operate for the next two flights, but after issues arose again prior to the 24 September flight, the decision was made to not power the MW/LW spectrometer for the duration of the deployment. Nevertheless, for the two intensive aircraft coordinations on 14 and 20 September, eMAS operated with both the SW and MW/LW spectrometers obtaining data. The flight tracks without the MW/LW spectrometer are shown in yellow in Fig. 1.

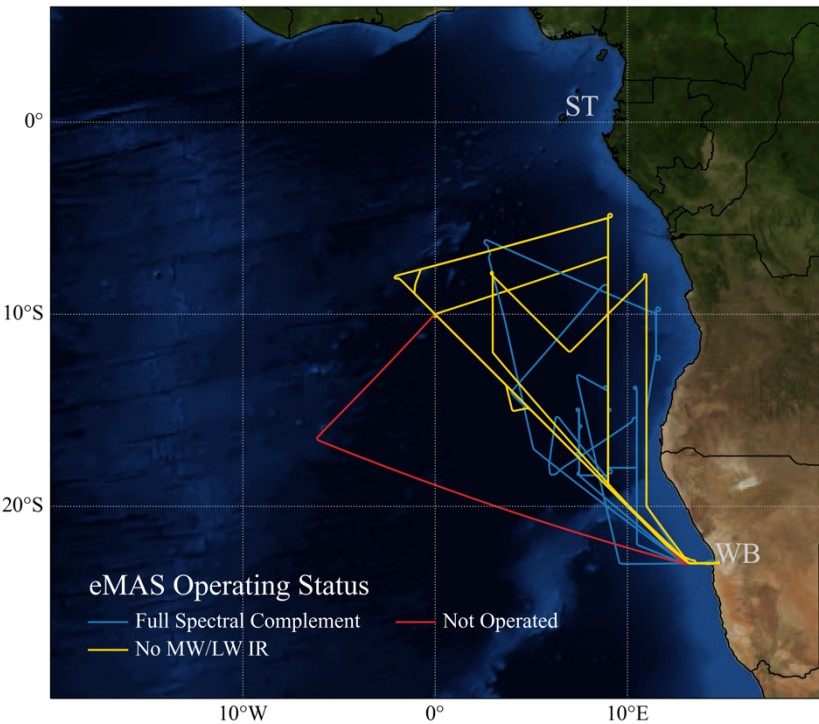


**Figure 1: The 10 ER-2 science flight tracks during ORACLES-2016 originating from Walvis Bay, Namibia (location indicated by "WB"). Tracks in blue are those for which the full complement of eMAS spectral channels was available (minus 7.3 µm). Tracks in gold are those for which the eMAS MW/LW spectrometer was not powered. The red track indicates the flight on which eMAS did not operate.**




## 3.2 eMAS Radiometric Calibration

Science analysis of eMAS field campaign data, in particular generating L2 geophysical retrievals for use by the science team and the broader community, relies on the ability to characterize with some degree of confidence the absolute radiometric calibration of both the SW and MW/LW spectrometers. For the MW/LW spectral channels, in-flight radiometric calibration is
monitored by observing two IR blackbody sources once each scan.

Radiometric calibration of the SW spectral channels, on the other hand, is a more onerous process. It includes pre- and post-deployment laboratory calibration using ASF's NIST traceable standard integrating hemispheres and in-field radiometric stability monitoring using a stable portable integrating hemisphere prior to each science flight. However, the eMAS optical
path is exposed to ambient conditions in-flight, thus the applicability at flight altitude of the radiometric calibration determined at ground level is uncertain. These ground monitoring activities therefore only provide insight on relative calibration changes and trends that often occur during the campaign due to the accumulation of dust, dirt, or other debris on the optics that can gradually degrade the radiometric response over time. They are not used to establish absolute radiometric calibration. Absolute radiometric calibration instead is established via comparisons of cloudy reflectance and cloud optical/microphysical property
retrievals with those of co-located space-borne imagers during targeted satellite under flights and, when available, via reflectance comparisons against vicarious calibration sites (Bruegge et al., 2021; Hook et al., 2001; King et al., 2010) obtained from overflights of such sites before, during, and/or after the campaign.

For ORACLES-2016, the pre- and post-deployment laboratory calibration and in-field hemisphere data suggested a non-
negligible degradation (darkening) of the SW channels with time, though flight-to-flight variability of the hemisphere data precluded definitively quantifying the change. Thus, for the first time, we used co-located RSP total reflectance observations to establish the radiometric degradation trend, and flight-to-flight changes to the calibration adjustments, in the primary spectral channels used for cloud optical and microphysical property retrievals. For the absolute calibration, three coordinated under flights of Aqua MODIS were obtained, occurring on 18, 20, and 27 September, though broken cloud conditions on 27
September precluded its use in the calibration analysis.

Our comparisons with Aqua MODIS during the satellite under flights, and with RSP during all science flights, suggest that eMAS experienced a roughly 5-7% radiometric degradation in its SW channels over the course of the campaign. Moreover, this degradation was not linear with respect to flight number as had been assumed in past campaigns. Figure 2 shows the
temporal radiometric adjustment factors (solid lines), derived from the RSP comparisons, for the six primary eMAS channels used for cloud optical and microphysical property retrievals, plotted as a function of flight date. The calibration results from the Aqua MODIS comparisons, used to anchor the RSP flight-to-flight changes to an absolute calibration benchmark (i.e.,



shifting the RSP results up or down based on the comparisons with MODIS), also are shown for the 18 and 20 September under flights (dashed lines and boxes).

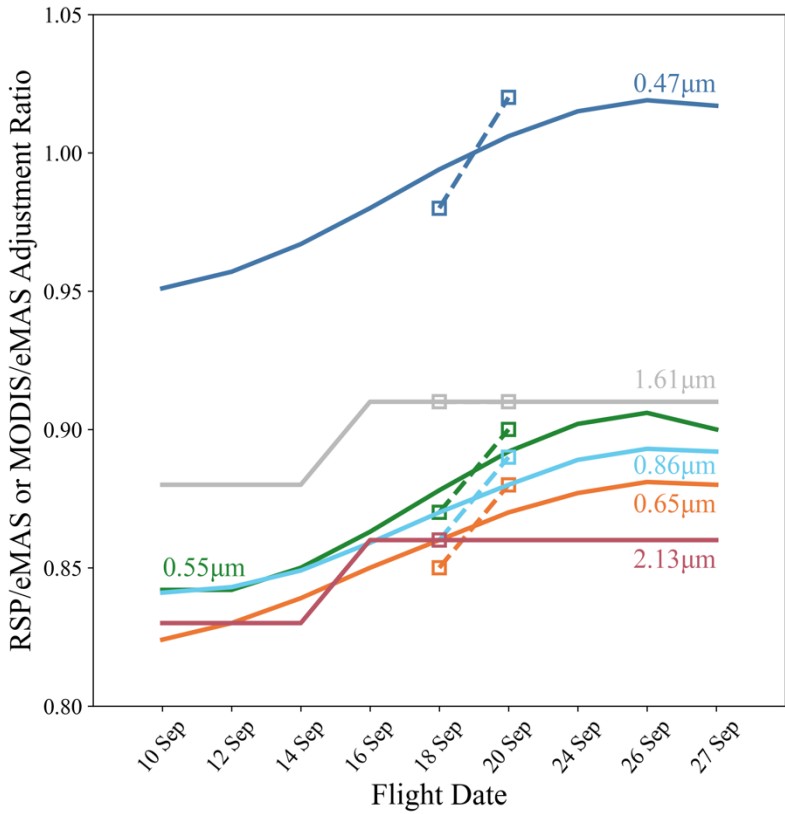


**Figure 2: Flight-dependent radiometric adjustment factors applied to the eMAS shortwave spectral channels. The solid lines, capturing flight-to-flight eMAS changes, were derived from comparisons with RSP (assumed stable across all flights), while the absolute magnitudes were anchored by comparisons with Aqua MODIS on 18 and 20 September (dashed lines with box symbols).**

Calibration of the remaining SW channels not included in Fig. 2, which do not have analogous MODIS counterparts and thus whose absolute calibration cannot be established directly from MODIS comparisons, instead relies on past vicarious calibration experience. Vicarious calibration data collected in 2013 after the SEAC4RS (Studies of Emissions and Atmospheric Composition, Clouds and Climate Coupling by Regional Surveys) campaign (Arnold et al., n.d.), and in 2019 after the FIREX-AQ (Fire Influence on Regional to Global Environments Experiment - Air Quality) campaign (Bruegge et al., 2021), showed 

a reasonably consistent channel-to-channel relative calibration. Given this consistency, we assume that the channel-to-channel relative calibration for ORACLES-2016 follows the 2013 and 2019 vicarious calibration results. Thus, we derive the calibration adjustments for the remaining SW channels relative to the six primary channels in Fig. 2 using the mean of the



2013 and 2019 channel-to-channel relative calibration offsets. Note that the eMAS L1B data in the public archive have the above adjustment factors pre-applied.

## 4 ORACLES Cloud Retrieval Results and Analysis

### 4.1 eMAS Standard Cloud Property Products

Example eMAS imagery and L2 cloud property retrievals from the ER-2 science flight on 14 September 2016 (flight track 4, 9:17-9:22 UTC) are shown in Fig. 3. Shown here are the true color RGB (0.47, 0.55, 0.65 µm) and retrieved cloud-top temperature (CTT), cloud optical thickness (COT), and cloud droplet effective radius (CER) from three spectral channels having heritage with MODIS (1.62, 2.13, 3.7 µm). The arrow in the RGB indicates the direction of flight of the ER-2, which for this track had a south-westward heading of 234.2° (relative to due north); the eMAS swath width and track length are denoted by the labels on the horizontal and vertical axes, respectively, of the RGB. As was often the case during ORACLES, only liquid phase boundary layer stratocumulus clouds were observed in this track. Note that the spectral CER retrievals exhibit differences that can be linked in part to real physics, namely differences in vertical penetration depths of spectral radiation (Platnick, 2000) coupled with droplet sizes that often correlate with height within the cloud consistent with the droplet growth of the adiabatic cloud parcel model. These differences are revisited in Sect. 4.2 via comparisons with in situ droplet size distribution measurements by the PDI and CAS/2D-S cloud probes.



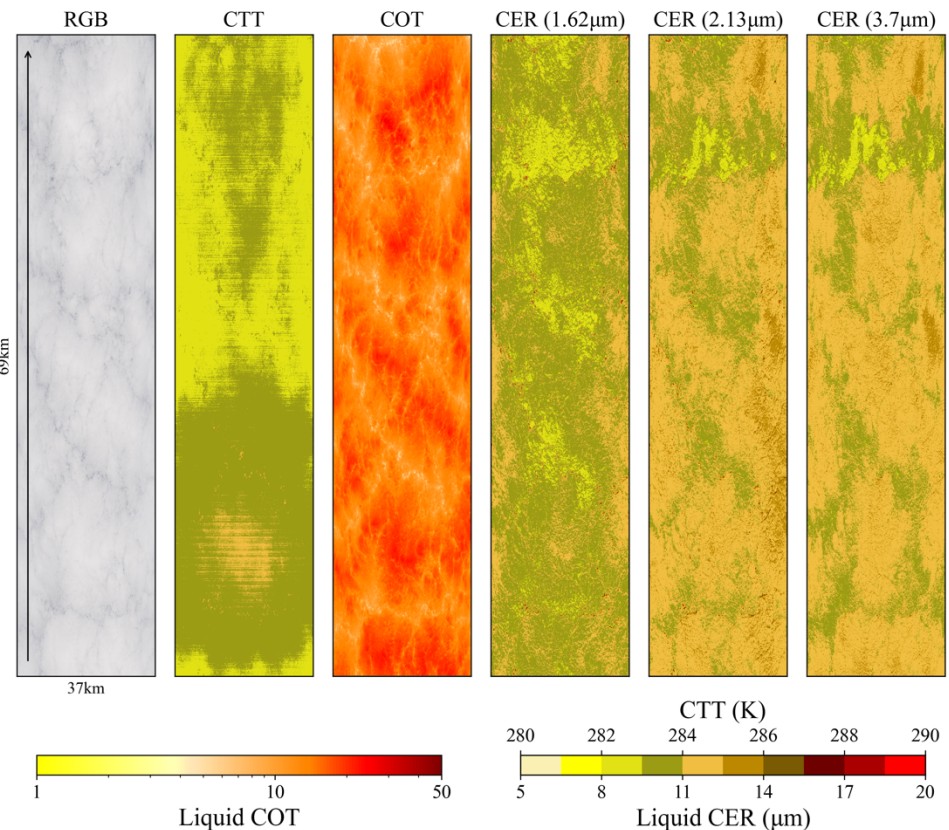

**Figure 3: Example eMAS imagery and standard Level-2 cloud property retrievals from the ER-2 science flight on 14 September 2016 (flight track 4, 9:17-9:22 UTC).**

Pixel-level retrieval uncertainties for the COT and spectral CER retrievals in Fig. 3 are shown in Fig. 4. Like their heritage MOD06 retrieval counterparts, these uncertainties account for known and quantifiable error sources such as instrument radiometry, atmospheric corrections (ancillary atmosphere profiles), surface spectral reflectance, and forward model errors (e.g., effective variance of the assumed droplet size distribution) (Platnick et al., 2017). However, unlike MOD06 that uses the pixel-level radiometric uncertainties reported in the MODIS L1B product, the eMAS L1B does not include pixel-level radiometric uncertainties and the cloud optical and microphysical retrievals must assume fixed relative radiometric uncertainties for retrieval uncertainty estimates; for ORACLES these radiometric uncertainties are set at 7% in all SW spectral channels and 5% for the solar component of the 3.7 μm MWIR channel. Uncertainties in the spectral CER retrievals are shown here to decrease with increasing wavelength, largely a consequence of the increasing orthogonality of the COT-CER solution space moving from 1.62 to 2.13 to 3.7μm that decreases sensitivity to measurement uncertainty (though the 3.7μm retrievals include additional error sources arising from the necessity to remove the contribution of thermal emission from the surface and atmosphere from the observed radiance).



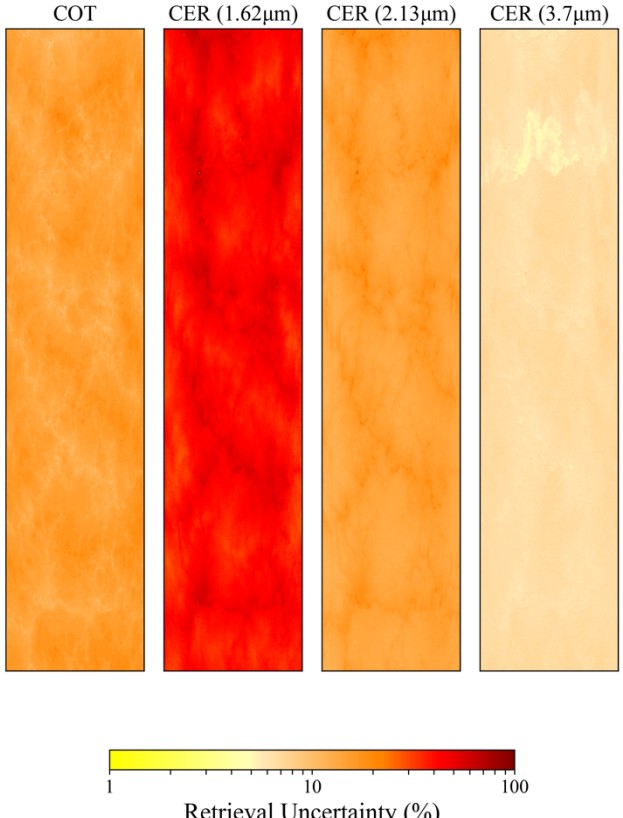

**Figure 4. Pixel-level retrieval uncertainties for the COT and spectral CER retrievals in Fig. 3.**


In addition to the MODIS-heritage 1.62, 2.13, and 3.7μm spectral channels, the eMAS spectral channel set includes four additional channels in the 1.6 and 2μm spectral regions (hereafter 1.6x μm and 2.x μm, respectively, when referring collectively to spectral channels in each region) that can be leveraged for CER retrievals, including channels analogous to the 2.25 μm

channel on VIIRS on NOAA's Suomi NPP and JPSS platforms and the 2.26 μm channel on RSP. The spectral response functions of the eMAS 1.6x and 2.x μm spectral channels (red lines), along with the spectral co-albedo for a notional liquid cloud (gray line; co-albedo defined as $1-\omega_0$, where $\omega_0$ is the single-scattering albedo), are shown in Fig. 5. Table 1 shows the channel center wavelengths for the 1.6x and 2.x μm spectral channels, and the 3.7 μm channel, along with band-averaged single-scattering properties ($\omega_0$, asymmetry parameter $g$, and extinction efficiency $Q_e$). The scattering properties in both Fig.

5 and Table 1 are computed for a liquid water cloud having CER = 10 μm.





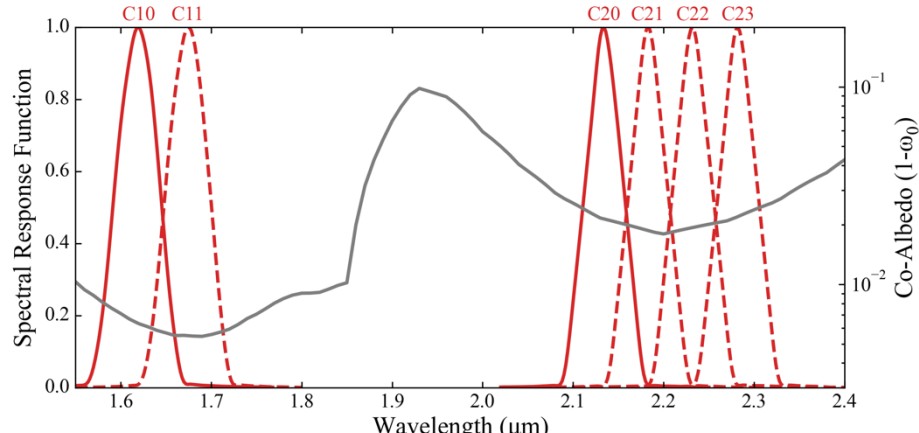

**Figure 5: Spectral response functions (red lines) for the eMAS SWIR channels in the 1.6x μm (channels 10, 11) and 2.x μm (channels 20-23) spectral regions. Also plotted is the co-albedo (gray line, defined as 1- $\omega_0$) for a liquid water cloud having CER = 10 μm.**

**Table 1: Single-scattering properties for the eMAS SWIR channels in the 1.6x and 2.x μm spectral regions, along with the 3.7 μm MWIR channel. These properties are calculated for a liquid cloud with CER = 10 μm. Note that the 1.62 μm (channel 10), 2.13 μm (channel 20), and 3.7 μm (channel 26) spectral channels are the MODIS heritage channels and are used for the standard eMAS CER retrievals.**

| eMAS Channel | Central Wavelength | $\omega_0$ | $g$ | $Q_e$ |
|---|---|---|---|---|
| | | *(liquid cloud, CER = 10 μm)* | | |
| 10 | 1.62 μm | 0.994 | 0.844 | 2.190 |
| 11 | 1.67 μm | 0.995 | 0.843 | 2.194 |
| 20 | 2.13 μm | 0.979 | 0.841 | 2.233 |
| 21 | 2.18 μm | 0.982 | 0.840 | 2.237 |
| 22 | 2.23 μm | 0.981 | 0.840 | 2.241 |
| 23 | 2.28 μm | 0.978 | 0.841 | 2.245 |
| 26 | 3.70 μm | 0.896 | 0.801 | 2.335 |

Figure 6 shows CER retrievals for the scene in Fig. 3 from the additional SWIR spectral channels along with those from the MODIS-heritage channels. It is interesting to note the deviation amongst these retrievals, not just between the 1.6x, 2.x, and 3.7 μm regions that has been noted in the past with MODIS, but also within the 2.x μm region that includes the various analogs to MODIS and VIIRS. A detailed analysis of these differences is shown in the microphysical retrieval evaluation in Sect. 4.2.





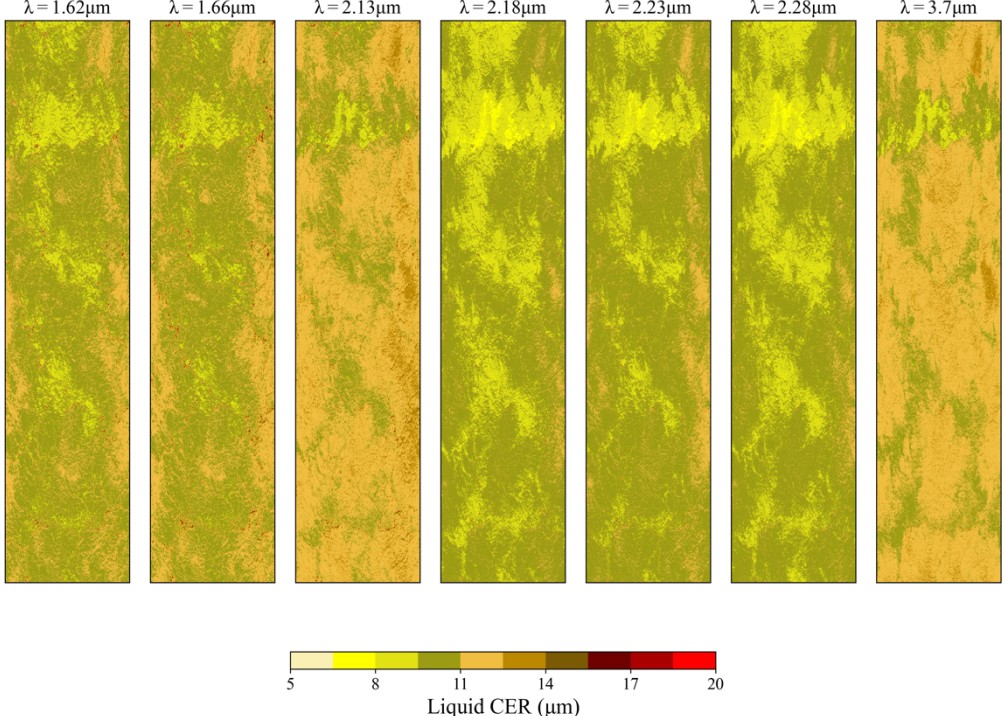

**Figure 6: Spectral CER retrievals from all available SWIR/MWIR eMAS channels (wavelengths from left to right: 1.62, 1.67, 2.13, 2.18, 2.23, 2.28, 3.7 μm) for the flight track in Fig. 3.**

An important caveat to consider regarding the eMAS cloud optical and microphysical property datasets is the impact of the spectral absorption of the aerosol layer overlying the MBL clouds in the SE Atlantic. Previous studies have shown that MODIS COT retrievals for the MBL stratocumulus clouds in the SE Atlantic can be biased low by up to 25% or more on a monthly mean scale due to above-cloud aerosol absorption in the 0.87 μm spectral channel (e.g., Meyer et al., 2013; Jethva et al., 2013; Alfaro-Contreras et al., 2014; Meyer et al., 2015). CER retrievals, on the other hand, are substantially less biased, e.g., less than 5% on a monthly mean scale (Meyer et al., 2015), since the above-cloud aerosol spectral absorption is at a minimum in the SWIR and MWIR (de Graaf et al., 2012; Haywood et al., 2004).

**4.2 Microphysical Retrieval Comparisons Against PDI, CAS/2D-S, and RSP**

During the 2016 ORACLES deployment, coordinated flights were performed that included ER-2 overpasses while the P-3 was sampling within the cloud layer. These co-locations of in situ cloud probe observations from the P-3 with the remote sensing observations of the ER-2 provide an opportunity to evaluate eMAS spectral CER retrievals against the CER derived from droplet size distributions measured by the probes. Given the spectral channel complement of eMAS that includes both MODIS and VIIRS analogs, this evaluation can inform our understanding of the differences in the sensitivities of the MODIS 2.13 μm



and VIIRS 2.25 µm SWIR channels to CER, a particular challenge for ongoing efforts towards cloud data record continuity between the two imagers (e.g., Platnick et al., 2020, 2021). We focus here on two coordinated maneuvers that occurred during the 20 September 2016 science flight.

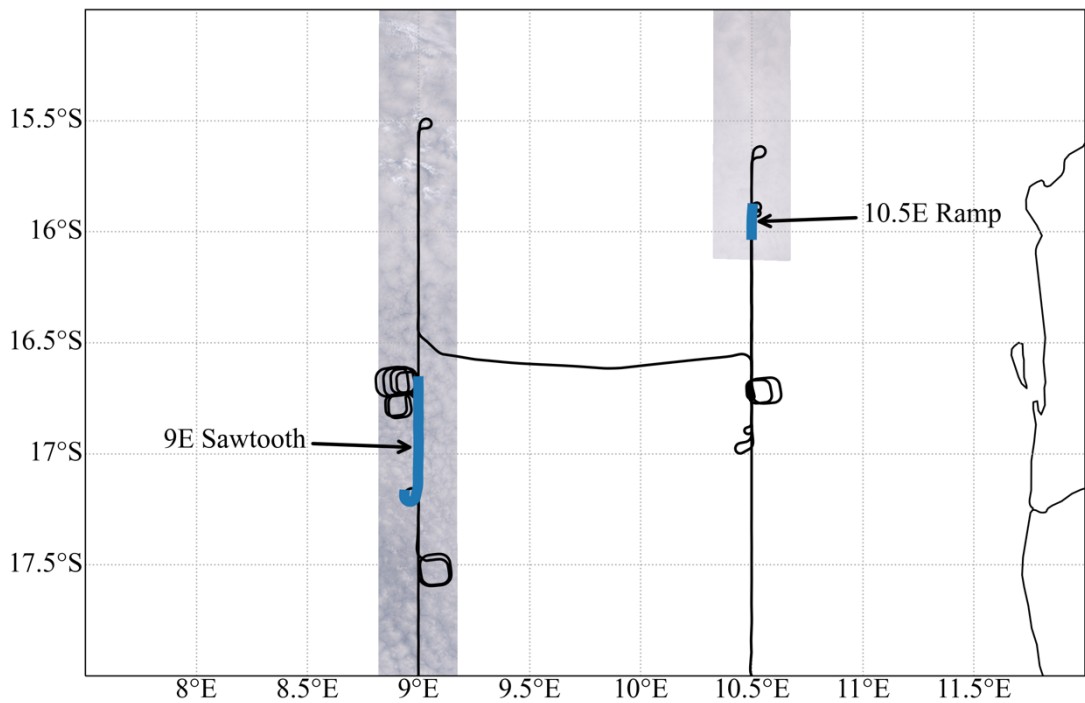

**Figure 7: P-3 flight track on 20 September 2016, featuring two coordinated maneuvers when an ER-2 overpass occurred while the P-3 was sampling within the cloud layer. True color RGB imagery obtained by eMAS during ER-2 overpasses are also shown. The locations of the two temporally coordinated maneuvers are indicated by the blue portions of the P-3 flight track. Each is designated by the P-3 flight module nomenclature used during ORACLES: a "Ramp" occurring at 10.5E longitude, and a "Sawtooth" within the cloud layer at 9E longitude.**


The 20 September P-3 flight track (black lines) in the region of two coordinated maneuvers with the ER-2, along with eMAS true color RGB imagery obtained during each coordination, is shown in Fig. 7. The specific locations where the P-3 was in the cloud layer while the ER-2 passed overhead, which serve as the case studies for this eMAS CER evaluation, are denoted by the blue segments. Each of these segments is labelled using the P-3 flight module nomenclature used during ORACLES: a

"Ramp" at 10.5E longitude (0937-0940 UTC) and a "Sawtooth" at 9E longitude (1142-1152 UTC), each part of broader "Radiation Wall" vertical flight tracks designed to sample the full depths of the cloud and aerosol layers along with the surrounding environment. The vertical profiles of the P-3 flight tracks during these coordinated Radiation Wall activities are shown in Fig. 8, with blue segments again denoting the location of ER-2 overpass while the P-3 was sampling within the cloud (note that the ER-2 made several overpasses at each location while the P-3 performed the Radiation Wall flight tracks shown).

During the 9E Sawtooth maneuver (Fig. 8 (a)), the P-3 sampled the full depth of the cloud layer repeatedly over a short





duration, providing good statistics on the vertical profile of the droplet size distribution. The 10.5E Ramp (Fig. 8 (b)), on the other hand, featured a single P-3 descent through the full depth of the cloud. While the P-3 also later ascended through the cloud layer in a stair-step fashion during this 10.5E Radiation Wall as the ER-2 again flew overhead, that portion of this coordination is excluded from the evaluation due to the uncertainties involved in piecing together a cloud vertical profile from
spatially decoupled layer sampling over a long distance.

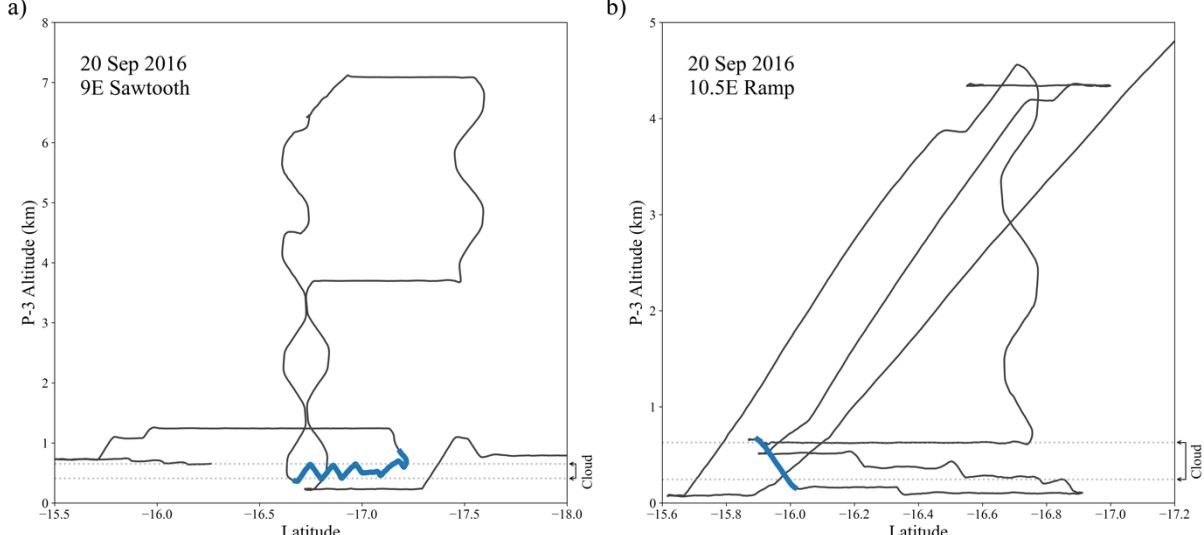

**Figure 8: Vertical profiles of the P-3 flight tracks during the two coordinated maneuvers with the ER-2 in Fig. 7. The blue segments denote the P-3 locations in the cloud while the ER-2 was overhead. (a) "Sawtooth" profile at 9E longitude where the P-3 repeatedly vertically sampled the full depth of the cloud layer over a short duration. (b) "Ramp" profile at 10.5E longitude where the P-3**
**descended through the cloud layer.**

Figure 9 shows eMAS imagery for the 9E Sawtooth coordination (from left to right): the true color RGB, observed scattering angle, COT, and CER from the 1.62, 2.13, and 3.7 μm channels, respectively. The RGB image also includes an arrow denoting direction of travel of the ER-2 and axis labels indicating the length and width of the scene. The blue outlined boxes in each
panel denote the approximate region in which the P-3 was sampling within the cloud during this ER-2 overpass (Figs. 7 and 8 (a)). This imagery shows relatively homogeneous, moderately optically thick (mean COT = 9.4 within the blue box) closed-cell stratocumulus clouds, a cloud regime that is expected to best conform to the plane-parallel radiative transfer assumptions used in the cloud optical property retrievals of eMAS and other airborne and spaceborne imagers. Note also that the spectral CER retrievals shown appear to conform to the adiabatic cloud assumption, with CER retrievals increasing with increasing
wavelength (and decreasing photon vertical penetration depth).



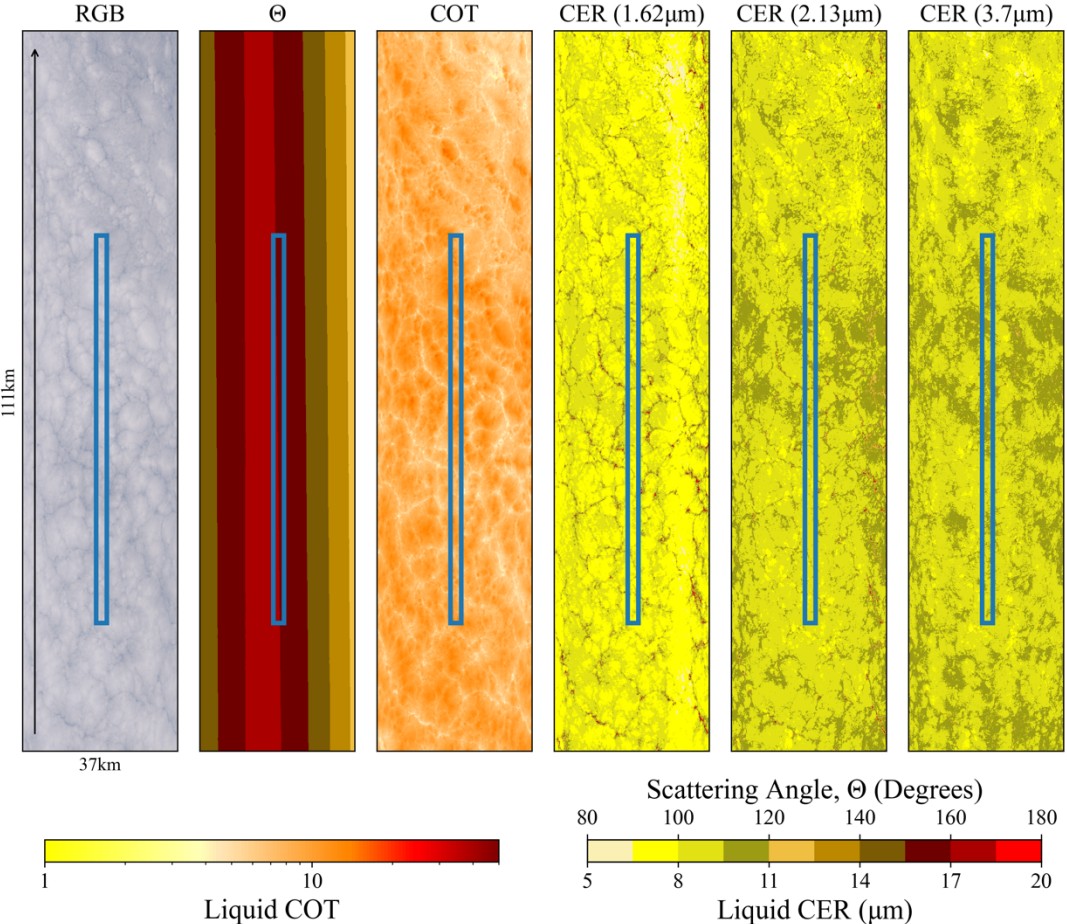

**Figure 9: eMAS imagery obtained during the 9E Sawtooth coordination shown in Figs. 7 and 8 (a). From left to right: true color RGB image, observed scattering angle (Θ), COT, and CER from the 1.62, 2.13, and 3.7 μm channels. The blue boxes in each panel denote the region where the P-3 was sampling within the cloud layer during the ER-2 overpass.**


CER profiles derived from PDI (blue) and CAS/2D-S (gray, red) DSDs obtained during the 9E Sawtooth are shown in Fig. 10 (a). The circles denote CER derived from the 1 s DSDs, while the triangles denote CER derived from the DSDs aggregated to fixed 10 m vertical layers. For CAS/2D-S, two profiles of CER are shown, one derived from the original DSDs (gray, denoted CAS for simplicity) and one derived from the DSDs having the CAS size bins shifted using the King LWC constraint (red,

denoted CAS$_{shifted}$). While there appears to be good correlation between CER from CAS/CAS$_{shifted}$ and PDI, the CAS$_{shifted}$ CER remain almost 2 μm smaller than PDI at cloud top, though these CER differences gradually decrease to near-zero at cloud base. Interestingly, the roughly 2 μm difference between PDI and CAS$_{shifted}$ at cloud top is within the range of MODIS CER retrieval biases found in previous comparisons against CAS and other legacy cloud probes (see (Witte et al., 2018) and references therein).






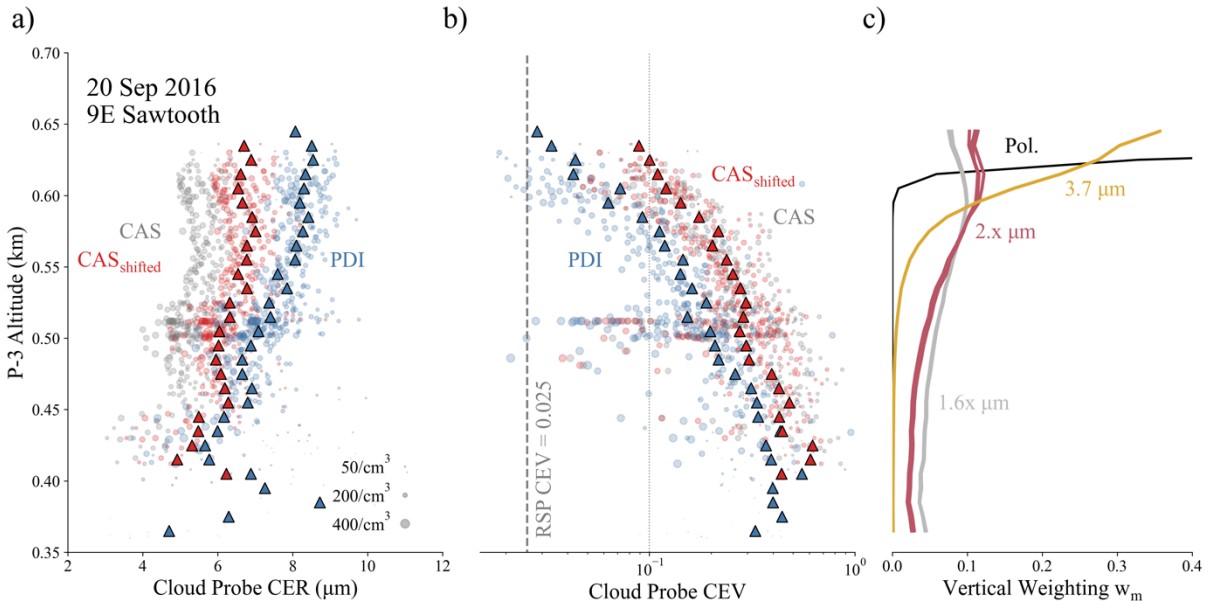

**Figure 10. P-3 cloud probe data obtained during the 9E Sawtooth coordination (blue segment in Fig. 8 (a) and corresponding blue box in Fig. 9), where PDI is plotted in blue, CAS/2D-S is plotted in gray, and CAS/2D-S with CAS shifted to correct for liquid water content biases is plotted in red. (a) CER profiles derived from the reported 1-second probe DSD observations (circles) and from DSD observations aggregated to 10 m vertical layers (triangles). (b) Profiles of probe DSD effective variance, CEV, for the 1-second observations (circles) and 10 m vertical layer aggregations (triangles). Also shown is the mean CEV retrieved from co-located RSP polarimetric observations (vertical dashed line) along with a vertical dotted line denoting the CEV assumed for the eMAS and RSP bi-spectral retrievals (CEV = 0.1). (c) Vertical weighting functions for the eMAS 1.6x μm (gray), 2.x μm (red), and 3.7 μm (yellow) channels, along with the single-scattering weighting function approximating polarization sensitivity (black), derived from forward radiative transfer calculations using the 10 m binned PDI DSD observations.**

Profiles of CEV computed from the PDI and CAS/2D-S (again denoted CAS and CAS$_{shifted}$ for simplicity) DSDs using Equation 7 are shown in Fig. 10 (b). Also shown in this plot is the mean CEV from co-located RSP polarimetric cloud-bow retrievals (vertical dashed line at CEV = 0.025) and the CEV assumed in the eMAS (and MODIS heritage) cloud optical property retrievals (vertical dotted line at CEV = 0.1). Both PDI and CAS$_{shifted}$ indicate a strong vertical CEV gradient, with CEV decreasing rapidly with increasing altitude. The two probes strongly disagree on CEV at the top of the cloud, however, with CAS$_{shifted}$ indicating CEV at cloud top roughly consistent with the heritage bi-spectral retrieval assumption and PDI indicating CEV roughly a factor of four smaller than CAS$_{shifted}$ and more consistent with the RSP polarimetric retrieval. This difference is notable given the sensitivity of bi-spectral CER retrievals to assumptions on CEV, and its implications are explored below.

Figure 10 (c) shows vertical weighting functions for the eMAS 1.6x (gray lines), 2.x (red lines), and 3.7 μm (yellow line) spectral channels, along with the RSP polarimetric weighting function (black line) approximated from a single-scattering




assumption (Miller et al., 2016), computed from the PDI 10 m DSDs used to compute the layer CER (triangles) in Fig. 10 (a)
and forward RT calculations following Equations 2-5. Consistent with (Platnick, 2000), the 1.6x µm channels have sensitivities deepest into the cloud with respect to cloud top, followed by the 2.x µm channels and finally the 3.7 µm channel that is sensitive only to the uppermost portions of the cloud.

**Figure 11: eMAS spectral CER retrieval statistics for the 9E Sawtooth coordination (blue box, Fig. 9) for the 1.6x (gray), 2.x (red),**
**and 3.7 µm (yellow) channels. Also shown are statistics of RSP multi-angle polarimetric cloud-bow retrievals (clear) and nadir spectral CER retrievals from both the GISS and GSFC algorithms (same colors as eMAS spectral regions). The dashed vertical lines in each panel denote the expected CER retrievals from each spectral region derived from the PDI and CAS/2D-S 10 m CER profiles**





and vertical weighting functions (Fig. 10). The four panels show retrieval statistics under different retrieval assumptions for eMAS and the GSFC RSP retrievals only: (a) MODIS heritage assumptions on CEV = 0.1 and liquid water complex index of refraction;

(b) MODIS heritage CEV assumption coupled with an updated liquid water complex imaginary refractive index measured at 295 K; (c) MODIS heritage refractive index assumption coupled with an updated assumption on CEV = 0.02, the latter roughly consistent with the co-located RSP CEV retrieval (Fig. 10); (d) updated assumptions on both the refractive index (295 K) and CEV (0.02). In panels (b)-(d), the dotted box/whiskers denote the MODIS heritage retrieval statistics shown in panel (a), highlighting the impact of the changes to each retrieval assumption. Note that the updated refractive index assumption is not available for the eMAS 3.7 μm

channel, thus those spectral channel retrieval statistics in (b) and (d) include only the heritage retrievals; neither the refractive index nor CEV assumption updates are available for the GISS RSP retrievals.

Figure 11 shows statistics of CER retrievals from eMAS and RSP (polarimetric and nadir-only bi-spectral) for the region within the blue outlined box in Fig. 9 that corresponds to the 9E Sawtooth coordination. The statistics are shown as box-

whisker plots for each retrieval (spectral channel or approach used labelled at left), with box face colors corresponding to spectral regions consistent with Fig. 10 (c), i.e., gray for the 1.6x μm channels, red for the 2.x μm channels, and yellow for the 3.7 μm channel, along with white for the polarimetric cloud bow retrievals. Both the mean (triangle) and median (vertical line within the box) for each retrieval are shown. The width of each box extends from the first to the third quartile, and the whiskers extend to the farthest data point within 1.5 times the inter-quartile range from the box. For clean visual analysis, outliers

exceeding this whisker range are omitted. For the nadir-only bi-spectral RSP retrievals, statistics are shown from both the GISS and GSFC cloud retrieval algorithms, the latter statistics indicated by the inclusion of GSFC in the labels at left. Also shown, as the vertical lines spanning the extent of each panel, are estimates of expected eMAS spectral CER retrievals computed from the cloud probe 10 m CER profiles and vertical weighting functions (Fig. 10) using Equation 6, with the dashed and dotted lines being derived from PDI and CAS$_{shifted}$, respectively, and color indicating the spectral region consistent with

the box-whisker plots. Note that for both probes, only the expected CER specific to the 1.62, 2.13, and 3.7 μm channels are shown in the figure since the expected CER from the other 1.6x and 2.x μm channels are consistent within each spectral region, as shown in Table 2. Both the CAS$_{shifted}$ and PDI expected CER exhibit the well-known adiabatic signature, with CER(3.7 μm) > CER(2.x μm) > CER(1.6x μm), though for PDI all are within 0.6 μm of each other and for CAS$_{shifted}$ there is even less spread. Moreover, the CAS$_{shifted}$ CER are 1.4 to 1.6 μm smaller than the expected CER derived from PDI, a result that is roughly

consistent with the CER profiles shown in Fig. 10 (a).

Table 2. The expected CER, weighted CEV, and weighted optical depth within the cloud (τ) for the eMAS SWIR and MWIR channels, along with the single-scattering weighting approximating polarization sensitivity, for the 20 September, 2016, 9E Sawtooth case study, computed from the PDI and CAS$_{shifted}$ probe data.

**9E Sawtooth Case Study**

| eMAS Central Wavelength | PDI | | | CAS/2D-S | | |
|---|---|---|---|---|---|---|
| | CER (μm) | CEV | τ | CER (μm) | CEV | τ |
| 1.62 μm | 7.9 | 0.129 | 5.7 | 6.5 | 0.238 | 4.1 |
| 1.67 μm | 7.9 | 0.130 | 5.8 | 6.5 | 0.237 | 4.0 |
| 2.13 μm | 8.0 | 0.113 | 5.1 | 6.6 | 0.219 | 3.7 |





| | | | | | | |
|---|---|---|---|---|---|---|
| 2.18 µm | 8.0 | 0.113 | 5.1 | 6.5 | 0.223 | 3.7 |
| 2.23 µm | 8.0 | 0.114 | 5.1 | 6.5 | 0.223 | 3.7 |
| 2.28 µm | 8.0 | 0.110 | 4.9 | 6.5 | 0.222 | 3.7 |
| 3.70 µm | 8.3 | 0.062 | 2.4 | 6.7 | 0.165 | 2.4 |
| Pol. | 8.4 | 0.035 | 0.5 | 6.7 | 0.100 | 0.7 |


Figure 11 (a) shows the CER statistics using heritage cloud forward radiative model assumptions in the bi-spectral retrieval LUTs. For eMAS and the GSFC RSP retrievals, those LUT assumptions are consistent with the assumptions made for the MODIS MOD06 cloud optical property retrievals, namely single-scattering properties computed from Mie calculations

assuming a homogeneous cloud having DSD CEV = 0.1 and wavelength dependent complex indices of refraction for liquid water obtained from (Hale and Querry, 1973) for wavelengths $\lambda < 1.0$ µm, (Palmer and Williams, 1974) for 1.0 µm $< \lambda < 2.6$ µm, and (Downing and Williams, 1975) for $\lambda > 3.5$ µm. Note that the same radiative model assumptions are made for the GISS RSP bi-spectral retrievals, though the liquid water complex indices of refraction are derived from a compilation by (Segelstein, 1981) that nevertheless is consistent with the refractive index datasets used in MOD06. It is interesting to note that, for this

relatively homogeneous cloud case, the CER retrievals, both bi-spectral and polarimetric, are in good agreement with each other and with the PDI expected CER, with the exception of the eMAS 2.13 µm and 3.7 µm channels and the RSP 1.59 µm channel retrievals that are larger by a micrometer or more.

CER statistics using various combinations of updated cloud radiative model assumptions in the eMAS and GSFC RSP bi-

spectral retrieval LUTs are shown in Fig. 11 (b)-(d), specifically (b) updated imaginary index of refraction datasets for the SWIR channels obtained from laboratory measurements at 295 K (Kou et al., 1993; Platnick et al., 2020), (c) heritage refractive index datasets (see previous paragraph) but with a narrower DSD CEV (0.02) more consistent with both the PDI DSDs at the top of the cloud and the RSP polarimetric CEV retrieval (Fig. 10 (b)), and (d) a combination of the 295 K SWIR refractive indices and CEV = 0.02. For each panel, retrieval statistics using the heritage cloud forward radiative model assumptions,

consistent with Fig. 11 (a), are also included as the dotted box-whisker plots, providing a visual reference for CER retrieval changes due to the updated assumptions. Note that the eMAS 3.7 µm retrievals in (b) and (d) do not have updated retrievals as the 295 K refractive index dataset does not extend into the MWIR.

For all combinations of updated cloud radiative model assumptions in Fig. 11, the bi-spectral CER retrievals from all spectral

channels decrease, with the largest impacts being in the eMAS 1.6x and 2.13 µm channels due to changing the CEV assumption from 0.1 to 0.02 (see panels (c), (d)). While that change seems to yield better agreement amongst the 2.x µm channels for both refractive index assumptions, it appears to overcorrect the eMAS 1.6x µm retrievals, at least with respect to the expected spectral differences implied by both the PDI and CAS$_{shifted}$ weighted CER (dashed and dotted vertical lines, respectively). A partial explanation may be found looking back to Fig. 10. For this case, both probes indicate a strong vertical gradient of DSD





CEV (Fig. 10 (b)), with the smallest values (narrowest distributions) at cloud top, increasing rapidly towards cloud base. Like the expected CER computed from the probe data using Equation 6, these CEV profiles, coupled with the vertical weighting functions in Fig. 10 (c), imply that, for this case, the vertical sensitivities of the 1.6x, 2.x, and 3.7 µm spectral channels are deeper in the cloud where CEV is larger. Indeed, using Equation 6 to compute a weighted CEV for each spectral channel, i.e., replacing CER($z$) with CEV($z$), shows that the CEV for all the spectral channels is closer to the heritage 0.1 assumption rather

than the PDI value at cloud top and the RSP polarimetric retrieval whose single scattering sensitivity inherently is weighted to cloud top. The weighted CEV for all eMAS SWIR/MWIR spectral channels for this case, along with the weighted optical depth within cloud indicating the level of vertical sensitivity, are shown in Table 2. Moreover, the 1.6x µm retrievals in this case are most strongly affected by the CEV assumption simply due to the greater degree of non-orthogonality of the COT/CER solution space and the location within that solution space of the reflectance observations, as shown by Fig. 12.

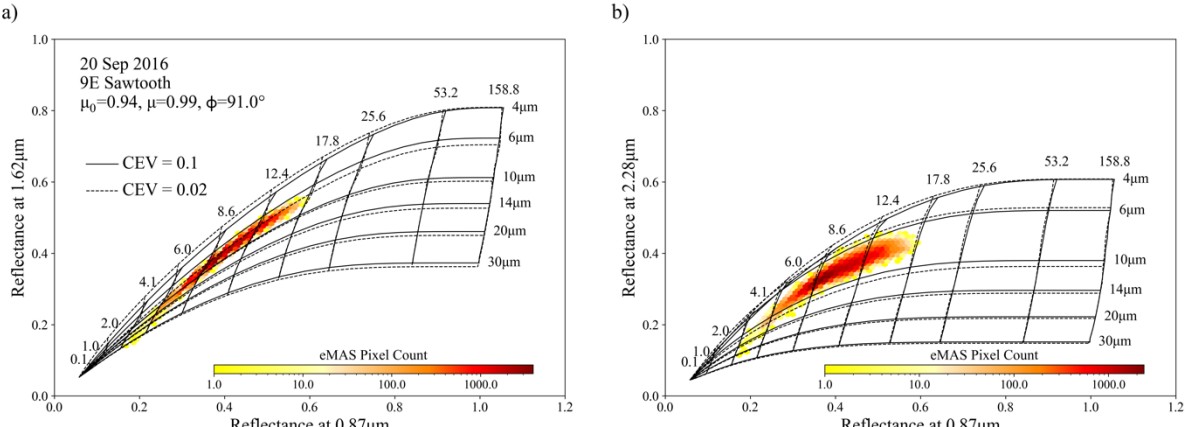


**Figure 12: COT/CER solution spaces for the eMAS (a) 0.87/1.62 µm and (b) 0.87/2.28 µm bi-spectral channel pairs. Solid lines indicate the heritage retrieval assumptions and dotted lines indicate the updated CEV assumption (i.e., Fig. 11 (c)). The distribution of actual eMAS spectral observations for the coordination box in Fig. 9 also are shown.**

eMAS imagery for the 10.5E Ramp coordination are shown in Fig. 13. Like Fig. 9, the blue outlined boxes in each panel denote the approximate region in which the P-3 was sampling within the cloud during this ER-2 overpass, which in this case was a single descent from cloud top to cloud base (Figs. 7 and 8 (b)), inherently over a much shorter distance with reduced sampling compared to the 9E Sawtooth case. Nevertheless, this imagery again shows relatively homogeneous stratocumulus clouds, though much more optically thick than the 9E Sawtooth case, with mean eMAS COT = 23.3 within the blue box.

Moreover, the spectral CER retrievals shown here appear to diverge from the adiabatic cloud assumption, with retrievals from the 2.13 µm channel yielding the largest CER.



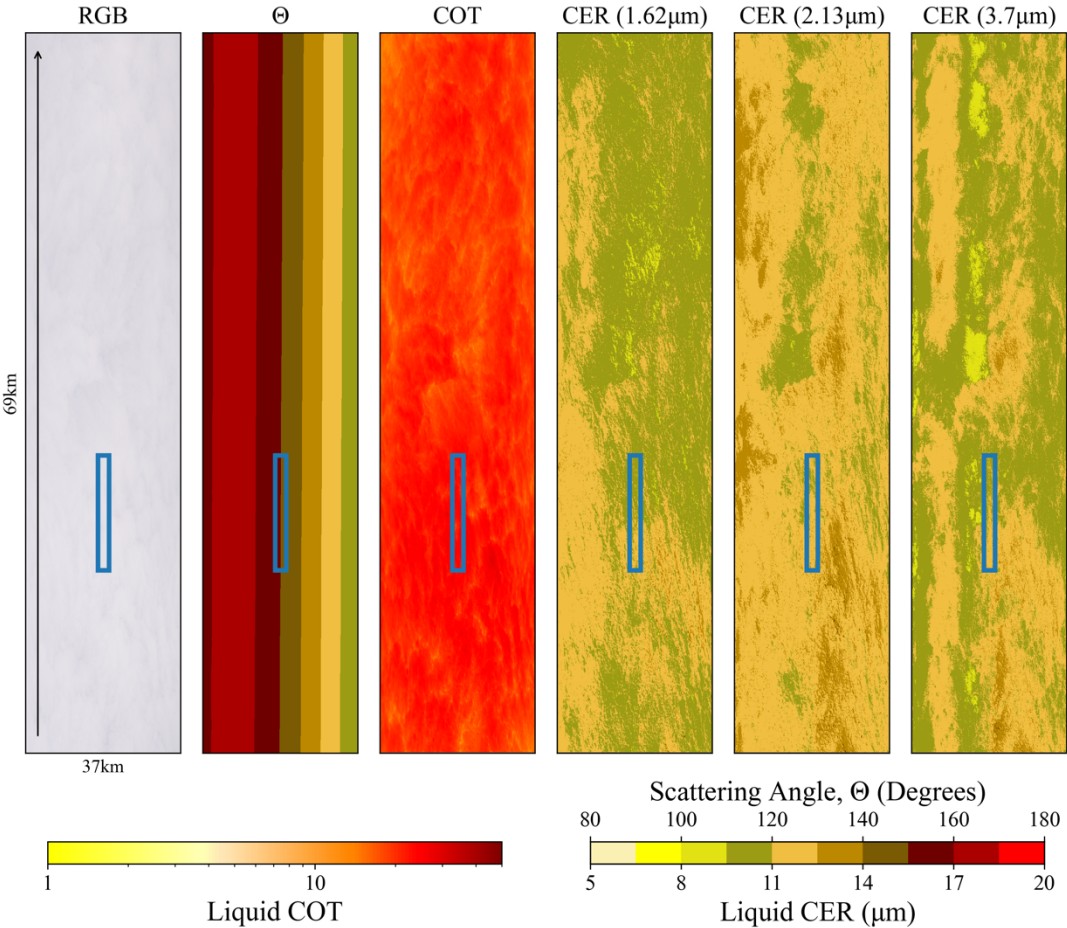

**Figure 13: eMAS imagery obtained during the 10.5E Ramp coordination shown in Figs. 7 and 8 (b). From left to right: true color RGB image, observed scattering angle (Θ), COT, and CER from the 1.62, 2.13, and 3.7 µm channels. The blue boxes in each panel**
**denote the region where the P-3 was sampling within the cloud layer during ER-2 overpass.**

The CER profiles derived from PDI (blue) and CAS/2D-S (gray, red) DSDs obtained during the 10.5E Ramp are shown in Fig. 14 (a). Like Fig. 10, the circles denote CER derived from the 1s DSDs and the triangles denote CER derived from the DSDs aggregated to the fixed 10 m vertical layers, and two CAS/2D-S profiles are shown, one derived from the original DSDs

(gray, denoted CAS for simplicity) and the other derived from DSDs having CAS size bins shifted given the King LWC constraint (red, denoted CAS_{shifted}). Again, there is good correlation between CAS/CAS_{shifted} and PDI, though CAS_{shifted} CER still are nearly 2 µm smaller than those derived from PDI at cloud top, with differences decreasing to near-zero at cloud base. Note, however, the obvious sampling reduction evident in the 1 s CERs compared with the 9E Sawtooth, due to the single descent through the cloud in this case.



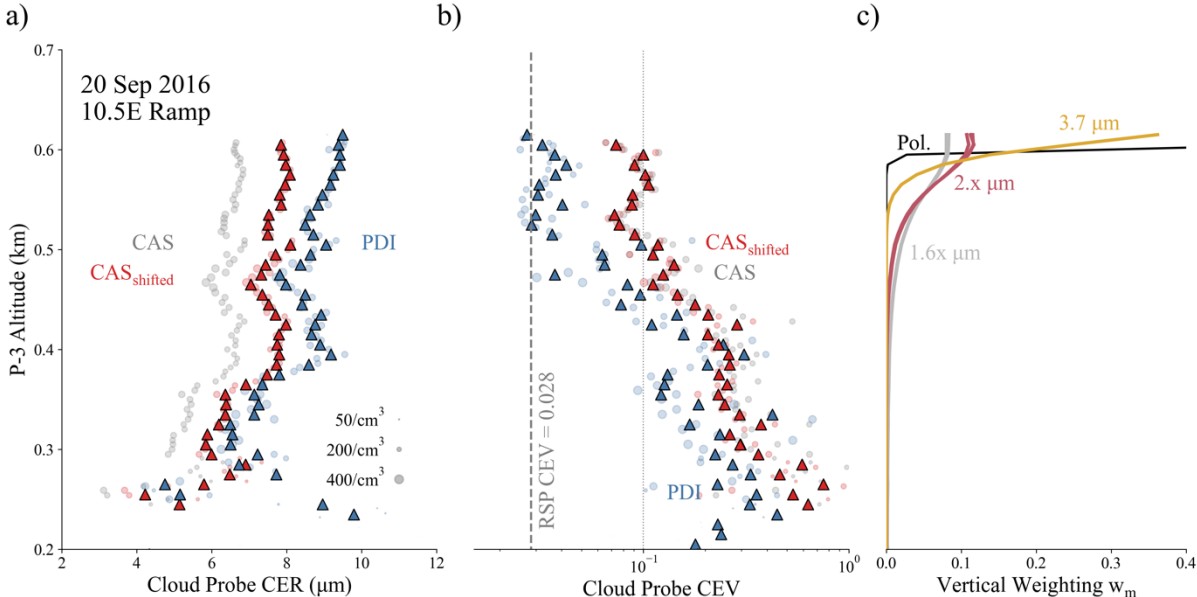


**Figure 14: P-3 cloud probe data obtained during the 10.5E Ramp coordination (blue segment in Fig. 8 (b) and corresponding blue box in Fig. 13), where PDI is plotted in blue, CAS/2D-S is plotted in gray, and CAS/2D-S with CAS shifted to correct for liquid water content biases is plotted in red. (a) CER profiles derived from the reported 1-second probe DSD observations (circles) and from DSD observations aggregated to 10 m vertical layers (triangles). (b) Profiles of probe DSD effective variance, CEV, for the 1-second observations (circles) and 10 m vertical layer aggregations (triangles). Also shown is the mean CEV retrieved from co-located RSP polarimetric observations (vertical dashed line) along with a vertical dotted line denoting the CEV assumed for the eMAS and RSP bi-spectral retrievals (CEV = 0.1). (c) Vertical weighting functions for the eMAS 1.6x (gray), 2.x (red), and 3.7 μm (yellow) channels, along with the single-scattering weighting function approximating polarization sensitivity (black), derived from forward radiative transfer calculations using the 10 m binned PDI DSD observations.**


The profiles of CEV computed from PDI and CAS/2D-S (again denoted CAS and $CAS_{shifted}$ for simplicity) DSDs are shown in Fig. 14 (b). The mean CEV from the co-located RSP polarimetric cloud-bow retrievals (vertical dashed line at CEV = 0.028) and the CEV assumed in the eMAS cloud optical property retrievals (vertical dotted line at CEV = 0.1) are shown also. As in the 9E Sawtooth case, there is a strong vertical CEV gradient throughout most of the cloud, except in the topmost 100 m or

so. The two probes again strongly disagree on CEV at cloud top, with $CAS_{shifted}$ indicating CEV roughly consistent with the heritage bi-spectral retrieval assumption and PDI indicating CEV roughly a factor of three smaller than $CAS_{shifted}$ and consistent with the RSP polarimetric retrieval.

The vertical weighting functions for the eMAS 1.6x (gray lines), 2.x (red lines), and 3.7 μm (yellow line) spectral channels,

along with the RSP polarimetric weighting function (black line) approximated from a single-scattering assumption, computed from the PDI 10 m DSDs in Fig. 14 (a) and forward RT calculations, are shown in Fig. 14 (c). Like the 9E Sawtooth case in Fig. 10 (c), the 3.7, 2.x, and 1.6x μm channels are each subsequently weighted deeper into the cloud. However, note that the




sensitivity for all spectral channels is practically limited to the topmost 100 m of the cloud where the CEV (Fig. 10 (b)) is roughly vertically invariant. The implications on retrieval sensitivities to CEV are discussed below.


**Figure 15: eMAS spectral CER retrieval statistics for the 10.5E Ramp coordination (blue box, Fig. 13) for the 1.6x (gray), 2.x (red), and 3.7 μm (yellow) channels. Also shown are statistics of RSP multi-angle cloud-bow polarimetric retrievals (clear) and nadir spectral CER retrievals (same colors as eMAS spectral regions). The dashed vertical lines denote the expected CER retrievals from each spectral region derived from PDI (Fig. 14).**






Figure 15 shows statistics of CER retrievals from eMAS and RSP for the region within the blue outlined box in Fig. 13 that corresponds to the 10.5E Ramp coordination. The retrievals and statistics shown, along with the color scheme, are consistent with the same plots for the 9E Sawtooth in Fig. 11. For the expected CER derived from the probes (vertical lines), again only

those specific to the 1.62, 2.13, and 3.7 µm channels are shown in the figure since the expected CER from the other 1.6x and 2.x µm channels are consistent within each spectral region, as shown in Table 3. Both the CAS$_{shifted}$ and PDI expected CER exhibit the well-known adiabatic signature, with CER(3.7 µm) > CER(2.x µm) > CER(1.6x µm), though with perhaps a weaker signature than in the 9E Sawtooth case for PDI where all are within 0.4 µm of each other. The differences between the PDI and CAS$_{shifted}$ expected CER shown in Table 3, ranging from 1.3 to 1.6 µm, also generally are smaller than what was seen in

the Sawtooth case, particularly for the 3.7 µm channel.

**Table 3. Same as Table 2 but for the 20 September 2016, 10.5E Ramp case study.**

**10.5E Ramp Case Study**

| eMAS Central Wavelength | PDI | | | CAS/2D-S | | |
|---|---|---|---|---|---|---|
| | CER (µm) | CEV | $\tau$ | CER (µm) | CEV | $\tau$ |
| 1.62 µm | 9.0 | 0.047 | 9.3 | 7.7 | 0.119 | 8.4 |
| 1.67 µm | 9.0 | 0.048 | 9.7 | 7.7 | 0.121 | 8.6 |
| 2.13 µm | 9.2 | 0.038 | 6.7 | 7.8 | 0.105 | 6.6 |
| 2.18 µm | 9.2 | 0.039 | 7.1 | 7.8 | 0.106 | 6.7 |
| 2.23 µm | 9.2 | 0.039 | 7.1 | 7.8 | 0.106 | 6.8 |
| 2.28 µm | 9.2 | 0.038 | 6.7 | 7.8 | 0.104 | 6.5 |
| 3.70 µm | 9.4 | 0.033 | 3.0 | 7.9 | 0.093 | 3.2 |
| Pol. | 9.5 | 0.028 | 1.0 | 7.9 | 0.079 | 1.2 |

Figure 15 (a) shows the CER statistics using the heritage cloud forward radiative model assumptions in the bi-spectral retrieval

LUTs. Unlike the retrievals for the 9E Sawtooth case in Fig. 11 (a), however, the bi-spectral and polarimetric retrievals here do not agree, with differences of 2 µm or more. Moreover, there is large disagreement amongst the bi-spectral retrievals themselves, up to 1.5 µm in the case of the eMAS 2.13 and 2.28 µm spectral channels. And none of the retrievals agree with the expected CER derived from the cloud probes, unlike the Sawtooth case where the retrievals were roughly consistent with PDI. Here, the polarimetric retrievals from RSP are smaller than the CER derived from PDI and the bi-spectral retrievals are

larger.

Figure 15 (b)-(d) show CER statistics using the same combinations of updated cloud radiative model assumptions in the eMAS and GSFC RSP bi-spectral retrieval LUTs as those used for the 9E Sawtooth case in Fig. 11 (b)-(d). Like the Sawtooth case, all combinations of updated radiative model assumptions yield smaller bi-spectral CER retrievals. The largest impacts result



from coupling the alternate refractive index assumption with the smaller CEV (again 0.02, more consistent with both the PDI
        DSDs at the top of the cloud and the RSP polarimetric CEV retrieval shown in Fig. 14 (b)), where, for instance, the 2.13 µm
        channel retrievals decrease by over a micrometer.

        Unlike the 9E Sawtooth case, however, the updated radiative model assumptions in the 10.5E Ramp case across the board
improve the bi-spectral retrieval agreement with the expected CER derived from PDI and move the bi-spectral retrievals closer
        to the RSP polarimetric retrievals. This is particularly the case for the combination of the 295 K refractive index and CEV =
        0.02 assumptions. Recalling the CEV profiles and vertical weighting functions in Fig. 14, all the spectral channels are sensitive
        primarily to the topmost 100 m of the cloud. While this physical vertical sensitivity is consistent with the 9E Sawtooth case
        (Fig. 10), in this 10.5E Ramp case the CEV is roughly vertically invariant within that portion of the cloud and, for PDI, is
consistent with the RSP polarimetric CEV retrievals, implying that the CEV experienced by the SWIR and MWIR spectral
        channels is closer to the RSP retrievals (CEV = 0.028) than to the heritage CEV = 0.1 assumption. This is confirmed by the
        computed weighted CEV shown in Table 3, where the spectral weighted CEV computed from PDI is between 0.035 and 0.05.
        It should be noted that the alternate CEV assumption is smaller than what the polarimetric retrievals and PDI suggest, implying
        the retrieval impacts shown here likely are overestimates. Nevertheless, using the exact polarimetric CEV, or the weighted
PDI CEV, still would decrease the bi-spectral CER retrievals and improve agreement with both PDI and the polarimetric CER
        retrievals. Moreover, it is interesting to note that the updated refractive index assumption further improves the bi-spectral CER
        retrieval agreement, both amongst themselves and with PDI. The same assumption in the 9E Sawtooth case (Fig. 11 (b)) also
        improved agreement amongst the bi-spectral retrievals while not adversely affecting their agreement with PDI and the
        polarimetric retrievals. The broader implications of the results of both the 9E Sawtooth and 10.5E Ramp comparisons are
discussed further in Sect. 5.

### 4.3 Retrievals at Glory Scattering Angles

        On multiple occasions during ORACLES 2016, the ER-2 flight heading was oriented such that eMAS observed within its
        swath scattering angles ($\Theta$) in the region around direct backscattering where liquid cloud spectral reflectance has distinct
        angular features. These backscatter angular features, known as the glory (Khare and Nussenzveig, 1977), occur at $\Theta$ roughly
between 160° and 180°, though the exact range depends on wavelength, with the NIR having the narrowest glory region and
        the MWIR the widest. The backscatter glory, much like the cloud bow observed in polarized NIR reflectance around $\Theta = 140°$
        (see, e.g., (Bréon and Doutriaux-Boucher, 2005; Bréon and Goloub, 1998)), is strongly coupled to the single scattering phase
        function and has similar sensitivities to the DSD of the cloud, with the location and relative amplitude of reflectance peaks at
        a given wavelength having sensitivity to CER and CEV, respectively (Spinhirne and Nakajima, 1994). Figure 16 illustrates
these glory sensitivities using liquid cloud single scattering phase functions in the backscatter region computed for the eMAS
        narrowband SWIR channel centered at 2.13 µm. Here, the normalized phase functions are shown for various (a) CER, assuming
        a fixed CEV of 0.1, and (b) CEV, assuming a fixed CER of 10 µm.





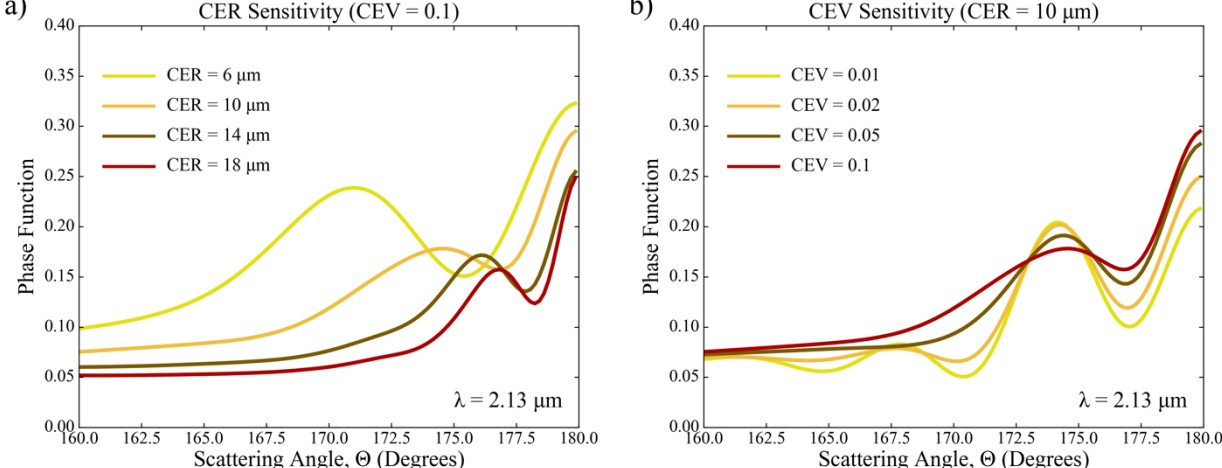

**Figure 16: Sensitivity of liquid cloud normalized phase functions in the glory scattering angle region to (a) CER and (b) CEV. Like**
**the polarized phase functions in the cloud-bow scattering angle region, the locations of the phase function peaks in the glory**
**scattering angle space are sensitive to CER while the amplitude of the peaks/valleys is sensitive to CEV.**

While the scattering angles sampled by satellite and airborne imagers are more frequently outside the direct backscatter region,
glories nevertheless often are observed in both airborne and satellite narrowband spectral reflectance observations and are most
readily apparent over relatively homogeneous liquid cloud fields such as marine stratocumulus. For instance, during the First
International Satellite Cloud Climatology Program Regional Experiment (FIRE) in 1987, glories over marine stratus clouds
were observed in NIR and SWIR spectral channel reflectance images obtained by two airborne cross-track scanning
radiometers flown aboard NASA's ER-2 aircraft, and extensive forward radiative transfer modeling was performed to
demonstrate the sensitivities of these glories to both CER and CEV (Spinhirne and Nakajima, 1994). Glories observed by
satellite imagers also have been used to infer DSD CEV (Benas et al., 2019) and cloud droplet size (Koren et al., 2022).

Of more relevance here, (Mayer et al., 2004) used airborne observations of the glory from the Compact Airborne
Spectrographic Imager (CASI) to simultaneously retrieve both CER and CEV for a stratocumulus case study using observed
reflectance in a single NIR spectral channel (0.753 μm). Like the polarimetric cloud bow retrieval, this glory retrieval finds
the CER and CEV pair whose simulated reflectance best matches the observed angular shape of the glory, i.e., matching the
angular location of the observed reflectance peaks to determine CER and matching the amplitude of the observed reflectance
peaks to determine CEV. While cloud reflectance also is a function of COT, the angular shape of the glory is assumed to be
nearly independent of COT and the retrieval least-square fitting is performed using the "glory reflectivity" that isolates the
shape of the glory by subtracting the mean reflectance of the scene (Mayer et al., 2004).




We perform a similar analysis here for the eMAS glory observations for the purpose of providing context for the eMAS bi-spectral CER retrievals and the above evaluation against RSP polarimetric retrievals and CER derived from in situ cloud probe DSDs, as well as for the CEV assumption used in the bi-spectral retrieval forward RT calculations. Following (Mayer et al., 2004), we define the "glory reflectivity" at eMAS spectral channel $b$ as a function of across-track pixel location $p$, such that


$$R_{glory,b}(p) = R_b(p) - \overline{R_b(p)}, \tag{7}$$

where $\overline{R_b(p)}$ is the mean reflectance across the eMAS swath. However, rather than implementing a rigorous least-square fitting approach to match the simulated glory reflectivity to the observed glory reflectivity, for our case study we simply infer CER

and CEV by visually comparing simulated reflectance peaks and amplitudes to the observations. In addition, we perform this analysis for three eMAS channels in the SWIR (1.62, 2.13 µm) and MWIR (3.7 µm) that have different vertical sensitivities within the cloud.

Figure 17 shows eMAS browse imagery for a flight track on 14 September 2016, where the backscattering region was observed.

The ER-2 direction of travel is indicated by the arrow at the left of the true color RGB; the track length also is indicated at left. While not visible in the true color RGB, the scattering angle (Θ) image clearly shows that eMAS observed scattering angles across the backscatter region in the right side of its swath. Moreover, the angular pattern of the glory is clearly visible in the retrieval imagery for both COT and spectral CER, indicating that the forward RT model calculations used for the retrieval inversion do not accurately describe the angular reflectance of the glory, a likely consequence of inadequate assumptions on

the DSD (e.g., CEV, vertical homogeneity).



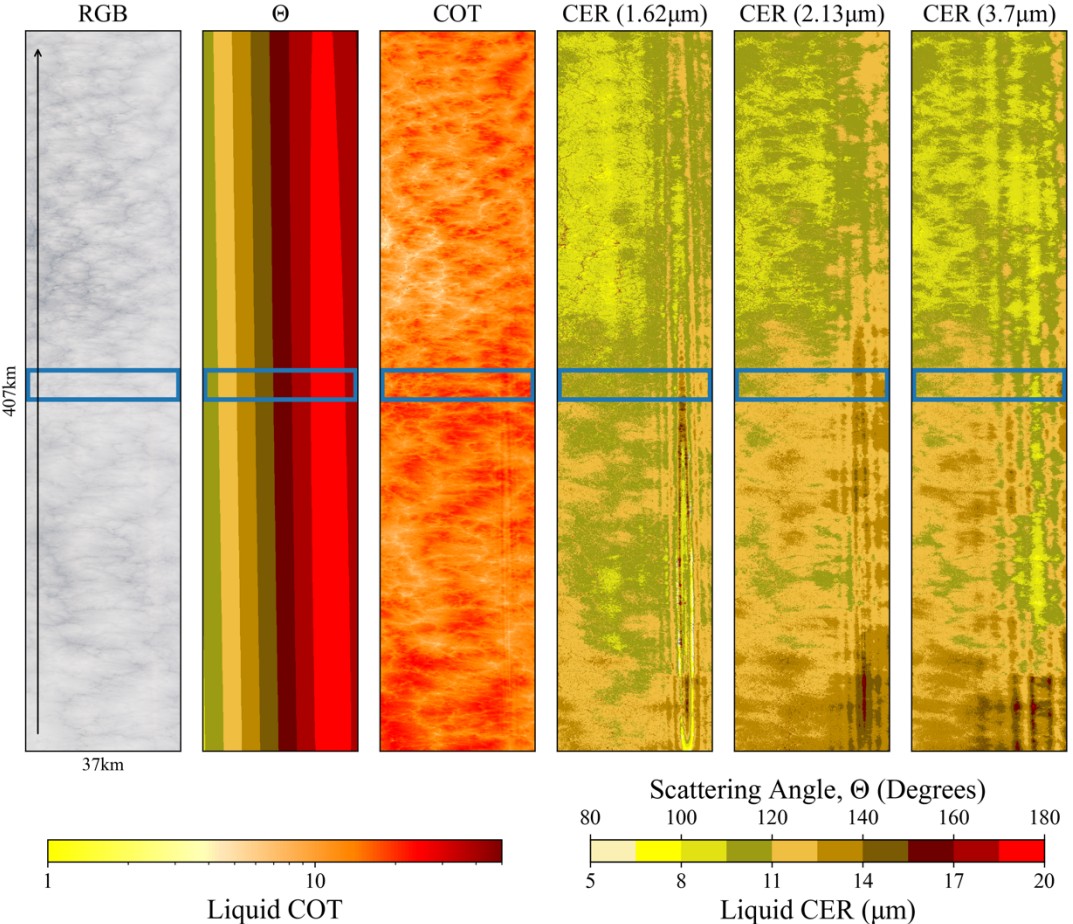

**Figure 17: eMAS browse imagery (true color RGB, scattering angle Θ, COT, and spectral CER from the 1.62, 2.13, and 3.7 μm channels) for an ER-2 flight track on 14 September 2016. ER-2 direction of travel is indicated by the arrow at the left of the RGB image; track length is also indicated at left. While the glory is not visible in the true color RGB using VIS wavelengths, its angular features are clearly visible in the retrieval imagery at right. The blue box denotes the region where the glory features in spectral reflectance are used to infer CER and v_eff.**

Figure 18 shows additional imagery and retrieval statistics for the blue outlined region in Fig. 17. A false color RGB using SWIR and VIS channels is shown in (a); the exact channel wavelengths used are denoted in the panel title at top. Note that we have stretched the color scale in this RGB such that the glory feature in the right side of the image is enhanced for visibility, though at the expense of also misleadingly intensifying the appearance of scene heterogeneity. The scattering angle observed by eMAS (black line), and the retrieved cloud-top pressure (CTP, blue line), is shown in (b), with the backscattering region (largest scattering angles) nicely correlating with the glory in (a). Across-track statistics for eMAS retrievals of COT (blue line) and spectral CER (black lines) are shown in (c). The lines here represent means of each retrieval computed for the along-track pixels at each across-track pixel location. eMAS spectral CER retrieval means are shown for the 1.62 μm (dotted black





line), 2.13 μm (solid black line), and 3.7 μm (dashed black line) channels. RSP polarimetric cloud-bow CER retrieval statistics for the same along-track region also are shown as the red box-whisker plot at the eMAS center pixel location, roughly where the RSP and eMAS observations are spatially co-located.

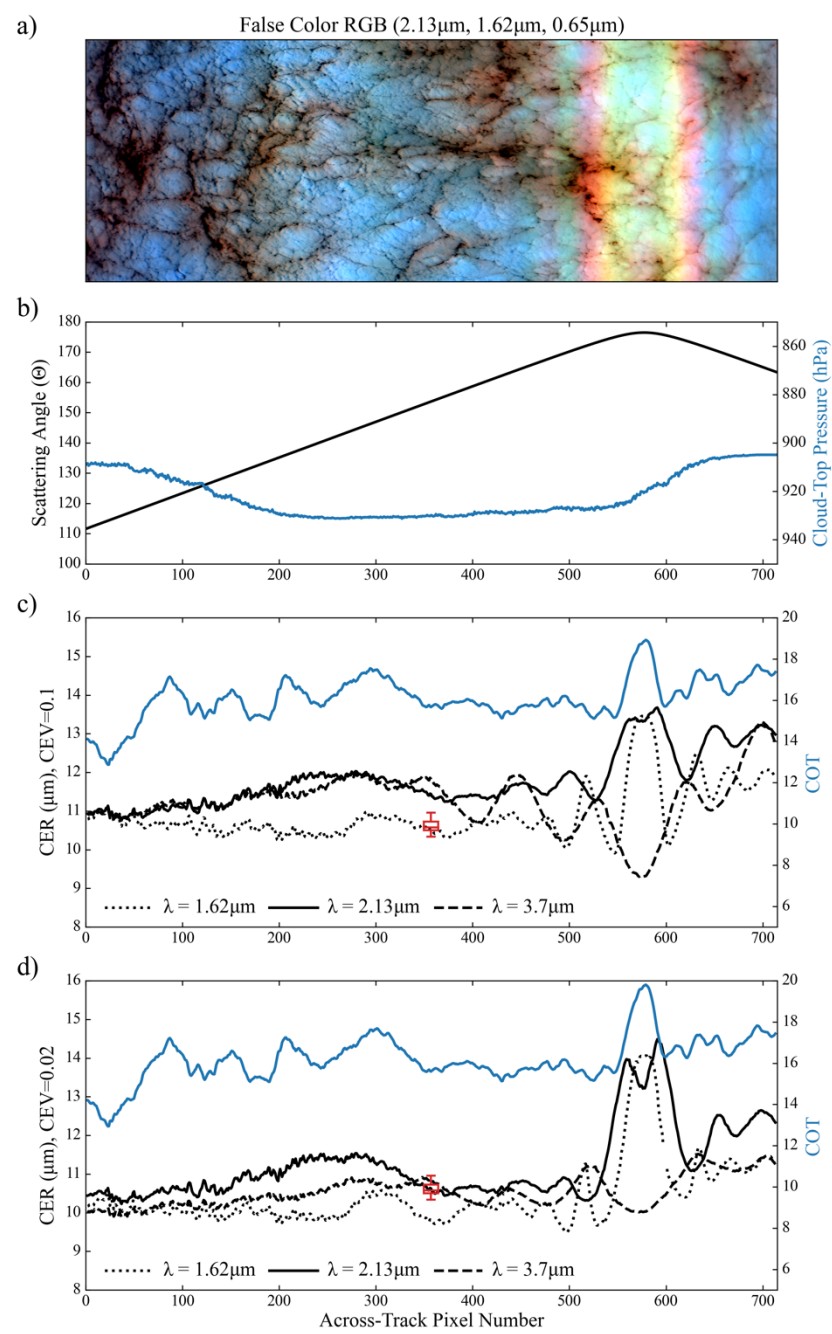

Figure 18: eMAS false color RGB (a) and across-track retrieval statistics for the blue outlined region in Fig. 17. The spectral channels used in the RGB (a) are indicated in the title at top. (b) The observed scattering angle (black line) and retrieved cloud-top pressure





(CTP, blue line). (c) Statistics for COT (blue line) and CER from the 1.62 μm (dotted black line), 2.13 μm (solid black line), and 3.7 μm (dashed black line) channels using heritage forward model assumptions (e.g., CEV = 0.1). RSP polarimetric CER retrieval statistics are shown as the red box-whisker plot at the center of the swath where RSP is roughly co-located with eMAS. (d) Same as (c) except assuming CEV = 0.02 in the eMAS forward RT model, an assumption that is close to the CEV = 0.017 retrieved by RSP near the swath center. The plots for all eMAS retrievals represent the along-track means of the retrievals at each across-track pixel location.

While the false color RGB in Fig. 18 (a) has the appearance of strong heterogeneity across the eMAS swath in this sub-region, the variability in the CTP and COT means across track is relatively small, though angular features, in particular a peak at the largest observed scattering angle, are evident for COT. Spectral CER retrievals, on the other hand, exhibit the strong angular patterns indicative of the glory across most of the righthand side of the swath – the 1.62 μm CER, for instance, has a fluctuation of more than 3 μm across the backscatter peak. These results are consistent with the retrieval sensitivities found by (Benas et al., 2019). Moreover, the angular peaks of each spectral CER retrieval across the glory region are not aligned, a consequence of the spectral dependence of the glory width that also produces the glory's distinctive rainbow-like appearance in the RGB. Interestingly, the RSP polarimetric cloud bow retrievals agree best with the eMAS 1.62 μm channel retrievals and are roughly 1.5-2 μm smaller than the 2.13 and 3.7 μm retrievals, consistent with the comparisons shown for the 20 September Sawtooth coordination in Fig. 11.

Plots of observed and simulated "glory reflectivity" $R_{glory}$ at cloud top for the glory portion of the scene in Fig. 18, specifically the rightmost portion of the swath starting at across-track pixel 450, are shown in Figs. 19, 20, and 21 for the 3.7, 2.13, and 1.62 μm channels, respectively. Note that the observed cloud-top $R_{glory}$ for each spectral channel (solid black lines) is computed from the eMAS reflectance observations corrected for above-cloud gaseous absorption and, like the retrieval statistics in Fig. 18, is the mean of the along-track pixel observations at each across-track pixel location. For the 3.7 μm channel, thermal emission also has been removed from the observations as part of the COT/CER retrieval process. For each plot in these figures, the rough locations of the observed $R_{glory}$ peaks are indicated by the vertical dotted black lines, providing a visual reference for inferring CER via $R_{glory}$ peak matching. The simulated $R_{glory}$ is computed for four values of CEV corresponding to the four panels in each figure, namely 0.1, 0.05, 0.02, and 0.01, moving from the top panel to the bottom panel. The CER values and corresponding line colors are indicated at the bottom of each figure and are computed for each CEV assumption.

Collectively, the $R_{glory}$ simulations in Figs. 19, 20, and 21 do not provide an unambiguous match for the observed $R_{glory}$ for any spectral channel, an unsurprising result when working with real observations. Nevertheless, some interesting conclusions can be drawn. Starting first with the 3.7 μm channel in Fig. 19, which is the easiest to decipher, there are three observed $R_{glory}$ peaks in this portion of the swath. Because eMAS does not observe the direct backscatter angles in this scene ($\Theta = 180°$ and angles immediately adjacent; see Fig. 18 (b)), the middle $R_{glory}$ peak is difficult to interpret and, given the simulated $R_{glory}$ shown, does not appear to exhibit sensitivity to CER beyond suggesting that CER likely is smaller than 12 μm. The two side peaks, however, do exhibit sensitivity to CER, as indicated by the shifting locations of the peaks in the simulated $R_{glory}$. Here,



the right side peak, near across-track pixel 650, implies CER of perhaps 11 μm given that the observed peak is roughly centered between the computed 10 μm (brown line) and 12 μm (red line) peaks. The left side peak near across-track pixel 500, on the
other hand, implies CER perhaps between 9-10 μm. This apparent increase in CER from the left side of the glory to the right side is consistent with the general increase in CER shown in the bi-spectral retrieval statistics in Fig. 18 (c), implying both approaches are characterizing real microphysical changes across the swath. In both sides of the glory, however, the CER are smaller than what the bi-spectral retrievals suggest.

The 2.13 and 1.62 μm channels in Figs. 20 and 21, respectively, tell much the same story even though the $R_{glory}$ peaks are increasingly more difficult to discern at shorter wavelengths. For the 2.13 μm channel, the primary left side $R_{glory}$ peak suggests a roughly 9 μm CER while the primary right side peak suggests a roughly 10μm CER. For 1.62 μm, only the right side primary peak is clearly discernible and suggests a CER of perhaps 9-10 μm. The tertiary peaks in the 1.62 μm $R_{glory}$, however, are more evident and suggest a roughly 9 μm CER on the left side and perhaps a 10 μm CER on the right. Similar to the 3.7 μm channel,
both the 1.62 and 2.13 μm channel glories suggest larger CER on the right side of the glory than the left, and both at least a micron smaller than their respective bi-spectral retrievals (Fig. 18 (c)). Moreover, the CER implied by the SWIR glories are smaller than those implied by the 3.7 μm channel glory.

For CEV, the 0.1 assumption (top panels in Figs. 19, 20, and 21) yields computed $R_{glory}$ for all three spectral channels that is
much too flat across the swath, without the strong angular features so evident in the observed $R_{glory}$. These angular features, however, become increasingly evident with decreasing CEV. While attempting to match the $R_{glory}$ peak amplitudes to an exact CEV is difficult given the additional sensitivity to COT, the simulations for all three spectral channels clearly suggest that the CEV for the vertical region of these clouds from which the single scattering glory signal originated is smaller than the 0.1 assumption used in the heritage bi-spectral multiple scattering retrievals. This result is supported, to the extent that CEV can
be assumed to not significantly vary across this scene, by RSP that retrieves, from the polarized cloud bow, CEV = 0.017 at the center of the swath in Fig. 18. eMAS COT and CER retrievals constrained by these CEV results, namely assuming CEV = 0.02 in the forward RT model, are shown in Fig. 18 (d). Compared to the CEV = 0.1 case in Fig. 18 (c), the bi-spectral retrievals are smaller and, except for the 1.62 μm channel, now are in much better agreement with the RSP polarimetric retrievals (red box-whisker plot) at the center of the swath. Moreover, the small scale oscillations around the backscatter peak in Fig. 18 (c)
have been reduced significantly. These results initially might suggest that the CEV = 0.02 forward model assumption is more appropriate for this scene than the heritage CEV = 0.1 assumption. However, the 1.62 and 2.13 μm channel CER retrievals at the largest scattering angles now are even larger than with the CEV = 0.1 assumption and are more out of family with the adjacent retrievals. These contradictory results imply that a single forward model assumption is unable to provide consistency across retrievals having differing information content and sensitivities to vertical heterogeneity, among others.



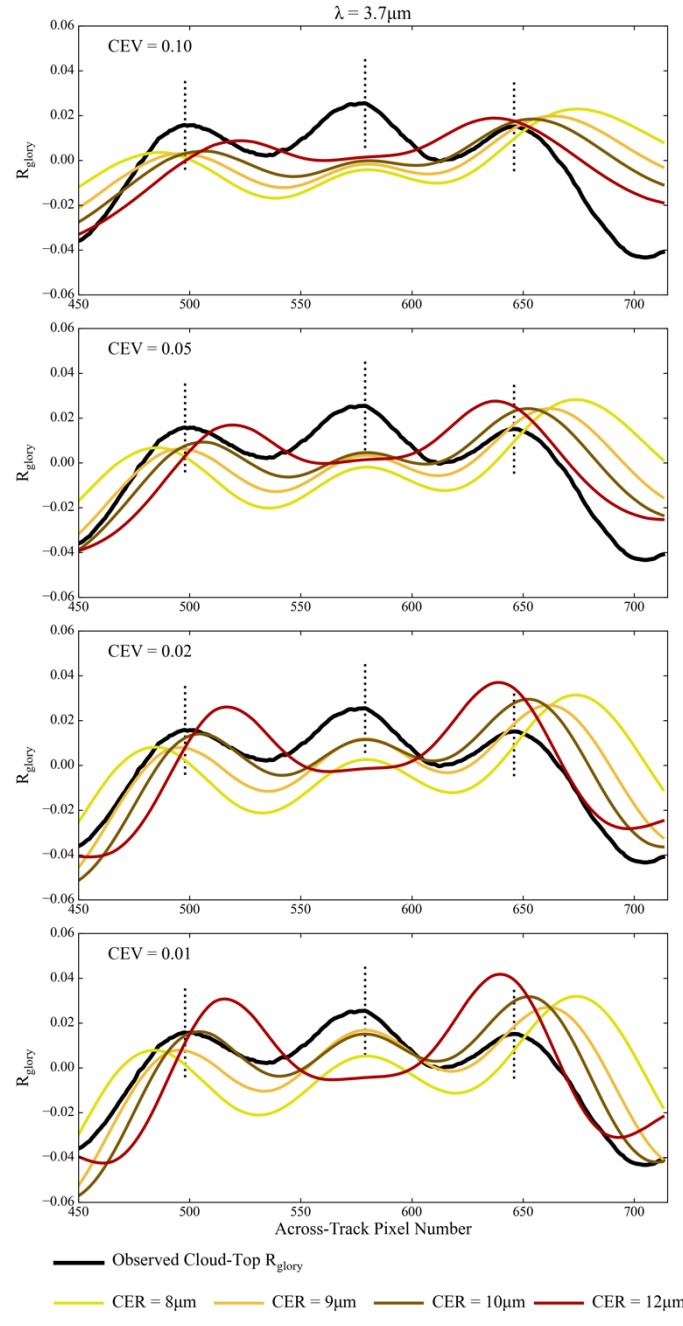

**Figure 19: Observed (black lines) and simulated "glory reflectivity" $R_{glory}$ (see Eq. 7 for definition) for the 3.7 µm eMAS channel corresponding to the right-hand side of the swath (starting at across-track pixel 450) in Fig. 18. The $R_{glory}$ simulations are performed for four CEV assumptions, namely 0.1, 0.05, 0.02, 0.01, moving from the top panel to the bottom panel, and for four CER as indicated by the legend at the bottom of the figure.**




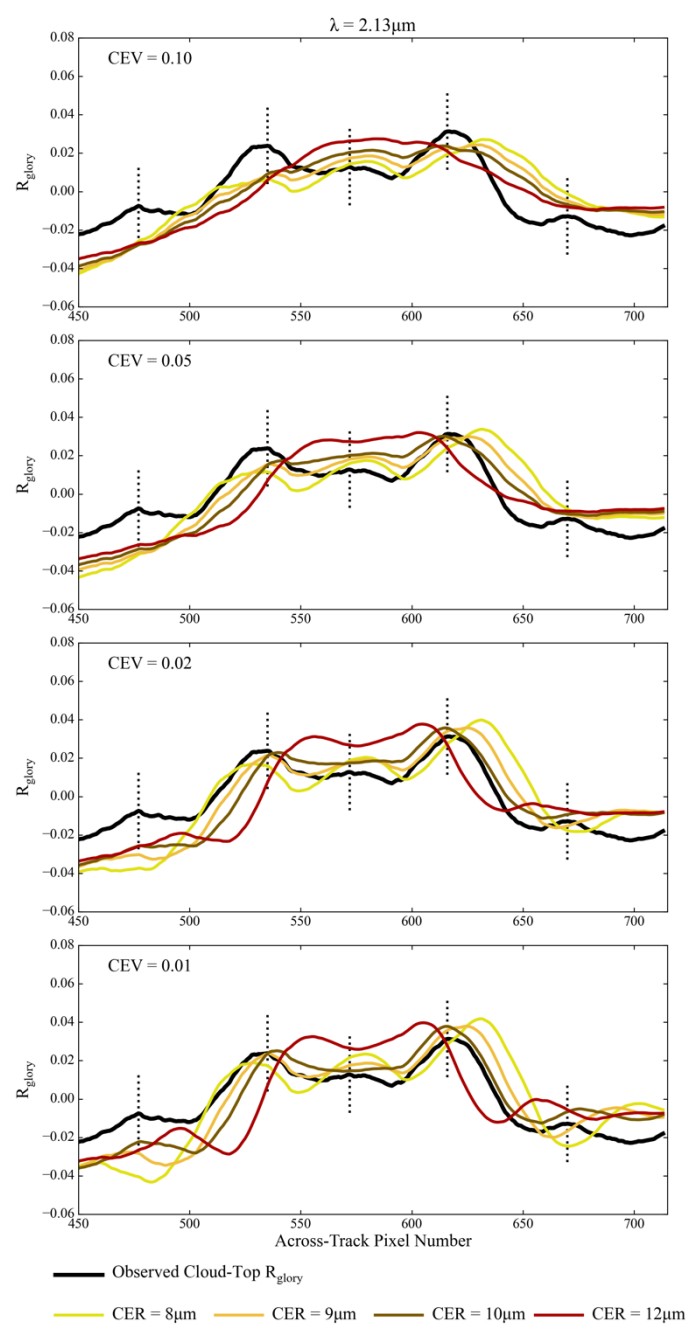


**Figure 20: Same as Fig. 19 but for the 2.13 μm eMAS channel.**





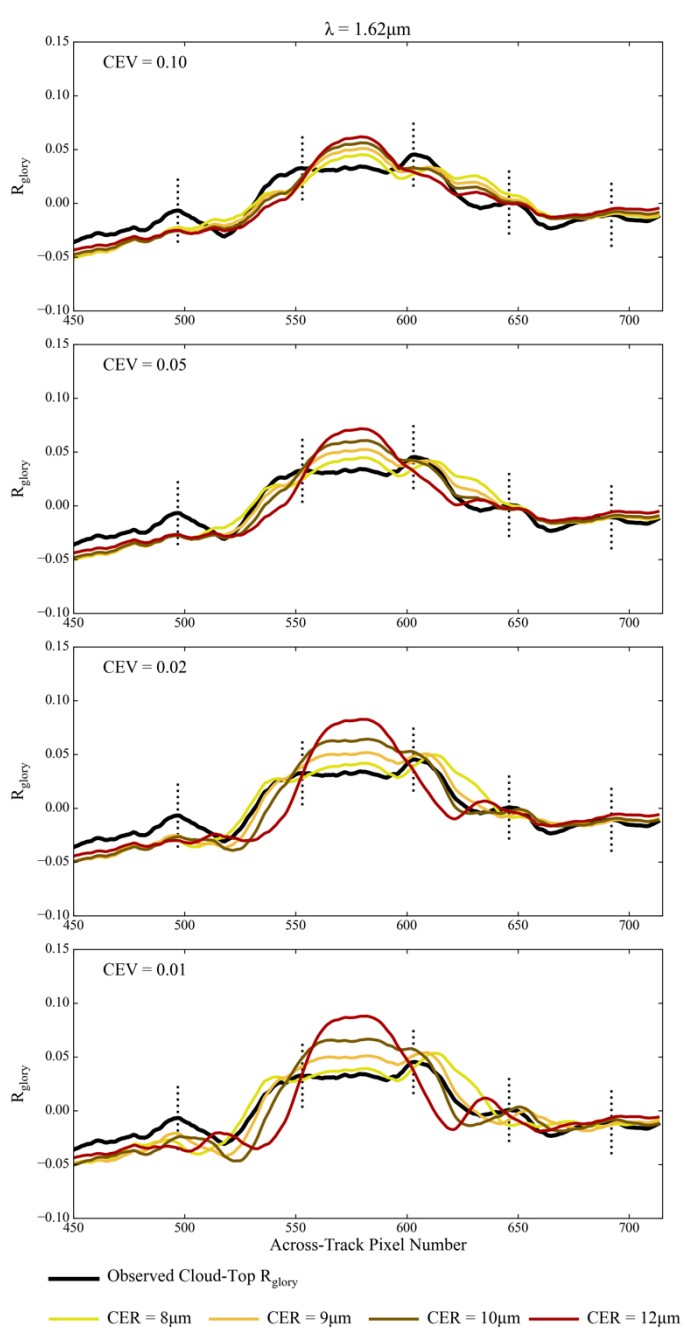

**Figure 21: Same as Fig.s 19 and 20 but for the 1.62 μm eMAS channel.**



## 5 Discussion


The results of the intercomparison in Sect. 4.2 and the glory analysis in Sect. 4.3 have important implications for interpreting existing spaceborne CER retrievals and their evaluations, and for informing remote sensing activities that seek synergy between spectral and polarimetric observations. Both analyses indicate that agreement amongst the various spectral and polarimetric retrievals is achieved in some cases and not in others. Furthermore, the use of alternate forward model assumptions

in the spectral retrievals, presumably more appropriate for the scenes in question, also yield mixed results. This is particularly the case when "constraining" the DSD CEV assumption used in the forward RT models for the spectral CER retrievals with CEV close to what was retrieved from RSP polarimetry. For example, in the two comparison case studies in Sect. 4.2, this polarimetric constraint helped retrieval agreement in the 10.5E Ramp case, where the cloud probes indicated weighted CEV closer to the RSP retrieval, but harmed agreement in the 9E Sawtooth case, where the probes indicated weighted CEV closer

to the heritage spectral retrieval assumption (CEV = 0.1). There is strong evidence from the co-located in situ cloud probe observations (Figs. 10 and 14, Tables 2 and 3) that these results are a direct consequence of the vertical profile of CEV in the cloud. This perhaps should not be surprising, since spectral cloud reflectance (intensity) and polarized cloud reflectance have sensitivities to different parts of the cloud. Spectral reflectance has strong contributions from multiple scattering that can extend deep into the cloud (see the vertical weighting functions in Figs. 10 and 14, particularly for the SWIR channels), while

polarization of the cloud reflectance is a single scattering phenomenon with contributions mainly from the very top of the cloud that may not be descriptive of the deeper cloud column that contributes to spectral reflectance (Miller et al., 2016, 2018). Thus, while leveraging complementary observations or retrievals as constraints on spectral retrievals is a worthy endeavor, for clouds with strong vertical heterogeneity the CEV retrieved from polarimetry at cloud top may be quite different from what is influencing the total spectral reflectance and may not be an appropriate constraint on total reflectance-based CER retrievals.

Such approaches must not be pursued cavalierly.

Considering the spectral retrievals themselves, their variability in the comparison case studies shown in Figs. 11 and 15 does not appear to conform to common conceptualizations of retrieval sensitivity, particularly the additional spectral channels beyond the heritage MODIS-like 1.62, 2.13, and 3.7 μm channels of eMAS. For instance, in the 9E Sawtooth comparison case,

both cloud probes indicate that CER increases with height, indicative of adiabatic droplet growth, a commonly observed, and assumed, characteristic of marine boundary layer stratocumulus clouds. Coupling this CER gradient with the spectral vertical sensitivities indicated by the weighting functions in Fig. 10 (c), we should expect an increase in retrieved CER moving from shorter to longer wavelengths, i.e., CER(1.6x μm) < CER(2.x μm) < CER(3.7 μm). Considering only the MODIS-like heritage 1.62, 2.13, and 3.7 μm eMAS channels, this retrieval pattern does indeed seem to hold. However, the additional 2.x μm

channels yield CER retrievals that are smaller than those from the 2.13 μm channel and are on the order of those from the 1.6x μm channels, inconsistent with the adiabatic assumption. This result implies that factors beyond vertical heterogeneity are at



play, such as radiometric calibration, above-cloud gaseous absorption correction errors, or forward RT model assumptions other than the DSD CEV discussed above.

- **Radiometric calibration:** As we describe in Sect. 3.2, for ORACLES the radiometric calibration of the additional 1.6x and 2.x µm eMAS channels is defined relative to the heritage MODIS-like 1.62 and 2.13 µm eMAS channels. The calibration of the heritage MODIS-like channels themselves is established via comparisons with co-located Aqua MODIS observations during targeted under flights during the campaign. While previous and subsequent in-flight vicarious calibration experience informed this approach, some uncertainty nevertheless is involved. Using the COT/CER solution space plots in Fig. 12 as a reference, calibration adjustments resulting in SWIR reflectances that are darker than they should be yield larger CER retrievals, and conversely adjustments resulting in SWIR reflectances that are too bright yield smaller CER retrievals. The CER retrievals from the additional 2.x µm channels in Fig. 11 thus might imply that those channels remain too bright due to underestimated calibration adjustments, though this is difficult to say with any certainty. Nevertheless, given the nature of airborne imager calibration, which for eMAS involves multi-sensor intercomparisons with relatively large uncertainties compared to laboratory or satellite on-orbit methodologies, calibration cannot be ruled out as a contributor.

- **Above-cloud gaseous absorption corrections:** While the spectral channels used for cloud optical property retrievals notionally are "window" channels, and indeed their locations are selected outside of absorption bands to minimize signal attenuation, no spectral channel enjoys a perfectly transparent atmosphere. For the SWIR channels used here, the absorption by three atmospheric constituents must be estimated and corrected for – water vapor for all channels, a $CO_2$ absorption band near 1.6 µm, and methane ($CH_4$) at wavelengths around 2.2 µm and longer. Errors in this atmospheric correction can arise from errors in the atmospheric profiles of the above constituents, from errors in cloud-top height retrievals that are used to define the above-cloud atmospheric column, or, to a lesser extent, from errors in the forward RT models used to compute atmospheric transmittance. The atmospheric corrections for the eMAS and GSFC RSP spectral retrievals are done via a MODTRAN-computed above-cloud transmittance LUT coupled with co-located atmospheric profiles from meteorological reanalysis data (Wind et al., 2020). We have evaluated the MODTRAN-based LUT approach against rigorous line-by-line calculations (LBLRTM (Clough et al., 1992)) for the comparison case studies in Sect. 4.2 and found good agreement between the two (results not shown), suggesting that the LUT approach is not an important source of retrieval error. Moreover, a sensitivity study (also not shown) using line-by-line calculations and doubled concentrations of $CO_2$ and $CH_4$ showed only small changes in above-cloud transmittance in the affected SWIR channels, thus implying only small impacts on spectral CER retrievals for more reasonable $CO_2$ and $CH_4$ concentration errors. For water vapor, on the other hand, which is the primary absorber across much of the SWIR, doubling its concentration in the two case studies in Sect. 4.2 yields decreased CER for all SWIR channels except 1.62 µm, as shown in Fig. 21. The biggest impacts are on retrievals from the 2.13 and 2.18 µm channels, with CER decreases of roughly 0.4-0.5 µm and 0.7 µm, respectively, for both case studies; impacts on CER retrievals from the other SWIR channels are roughly 0.3 µm or less. These CER



decreases are on the order of the impacts of the CEV and refractive index assumption changes shown in Sect. 4.2. However, assuming that a doubling of the concentration is a likely overestimate of the water vapor error, the resulting spectral CER retrieval errors can be expected to be smaller. Furthermore, the likelihood, sign, and magnitude of any such water vapor concentration errors cannot be quantified in practice such that it's difficult to determine the exact impacts on spectral CER retrievals for any given case study.

- **Forward RT model assumptions:** Exposing retrieval methodologies to new information content, such as new spectral channels, often reveals deficiencies in forward models and key assumptions. Such was the case for our previous experience extending MODIS cloud optical property retrieval algorithms to VIIRS for climate data record continuity, where large disagreement between CER retrievals from the MODIS 2.13 μm channel and the VIIRS 2.25 μm channel indicated an inadequate complex refractive index dataset for liquid water in the SWIR. An updated assumption on the imaginary index of refraction of liquid water using a dataset from more modern laboratory measurements made at 265 K (Kou et al., 1993) improved agreement in these retrievals (Platnick et al., 2020, 2021). For this analysis, in addition to the DSD CEV assumption discussed above, we also investigated the use of these alternate assumptions on the imaginary index of refraction of liquid water. While the 265 K measurements from the (Kou et al., 1993) dataset were chosen for the global application of the MODIS and VIIRS retrievals, that dataset also includes measurements at 295 K that are more appropriate for the stratocumulus clouds in the ORACLES region that, being located in the boundary layer, have warm cloud tops. This 295 K refractive index assumption did improve agreement amongst the SWIR retrievals in the comparison case studies in Sect. 4.2 when used alone, particularly for the 10.5E Ramp case (Fig. 15) but with only small impacts on the 9E Sawtooth case (Fig. 11). Coupling it with the CEV = 0.02 assumption appears to yield further improvement in the 10.5E Ramp case. These results are encouraging and merit further investigation, though the variability of the spectral retrievals still does not approach that expected from the co-located in situ cloud probes.



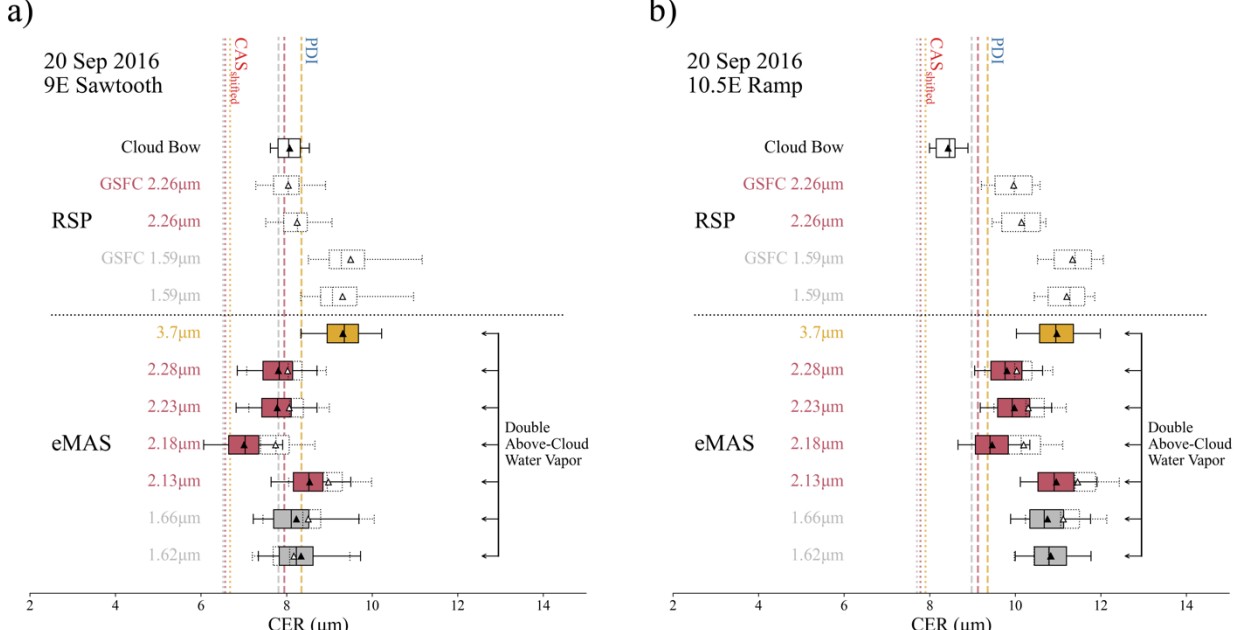

**Figure 21: The impacts on eMAS spectral CER retrievals due to doubling the above-cloud water vapor concentration in the atmospheric correction calculations. For all SWIR channels except 1.62 µm, such a doubling yields larger water vapor absorption (lower transmittance), larger atmospherically corrected cloud-top reflectance, and ultimately smaller retrieved CER.**

To complicate matters further, the in situ cloud probe measurements, used by numerous studies as a "ground truth" for evaluating imager retrievals of CER and attributing retrieval biases, also do not agree, with CER derived from PDI DSDs being 1.3-1.6 µm larger, and CEV roughly 50-60% smaller, than those derived from CAS/2D-S for the case studies shown here (see Tables 2 and 3). Previous studies have noted similar or larger differences between different probes, or like probes on different platforms, as context for remote sensing CER retrieval evaluation (King et al., 2013; Platnick and Valero, 1995; Witte et al., 2018). While some of these studies attribute such differences in part to errors associated with the probe measurements themselves (Witte et al., 2018), in general the errors and uncertainties of cloud probe DSD measurements perhaps are underappreciated. Propagating the stated bin-level sizing uncertainties for both PDI and CAS/2D-S (±0.5 µm) for the two case studies shown in Sect. 4.2 indicate that derived layer 1-sigma CER uncertainties are on the order of ±1 µm for both probes at the top of the cloud, as shown in Fig. 22 (a). These uncertainties are on the order of the CER differences between the probes as well as the spectral retrieval differences shown in the case study comparisons in Figs. 11 and 15. In addition, the bin size adjustment applied to CAS, derived from the King LWC constraint, yields derived CER increases of 1 µm or more (see Figs. 10 (a), 14 (a)). This again is on the order of, or exceeds, the spectral retrieval differences. Moreover, the representativeness of the probe data with respect to the broader FOV of the remote sensing instruments is inherently dictated by the sampling strategies within the cloud, the footprint of the remote sensing instrument, and the heterogeneity of the scene. Such potential



sampling biases are demonstrated in Fig. 22 (b) and (c) using the PDI CER from the 9E Sawtooth case in Fig. 10. In Fig. 22 (b), the P-3 altitude during the entire sawtooth maneuver is plotted in blue, with a single ramp of the sawtooth highlighted in

gray. CER from both the entire sawtooth (blue symbols) and the single ramp (gray symbols) are plotted in Fig. 22 (c) where, as in Fig. 10 (a), circles represent the 1 s observations and triangles represent the observations aggregated to 10 m vertical levels. The layer CER from the single ramp profile (gray triangles) is relatively consistent with the layer CER from the entire sawtooth (blue triangles) in the middle of the cloud, though near cloud top where the spectral retrievals have their sensitivity, the two profiles have a bias difference of 0.5 µm or more. These bias differences at the top of the cloud yield larger expected

spectral CER from the single profile (computed again using the weighting function methodology in Section 2.4) than from the full Sawtooth, roughly 0.25-0.26 µm larger for the 1.6x and 2.x µm channels and roughly 0.4 µm larger for the 3.7 µm channel. While these differences for this case are within the uncertainty of the PDI observations, the clouds sampled here are more spatially homogeneous than most clouds globally and these results likely cannot be extrapolated beyond this scene – for instance, to the 10.5E Ramp case in Figs. 13-15 and Table 3. In any event, probe sampling biases with respect to the remote

sensing FOV that may exist in any specific case study in practice cannot be quantified. Thus, taken collectively, the uncertainties and errors of the probe DSD measurements themselves and potential sampling biases within a given instrument FOV suggest that, while in situ cloud probes are useful tools for understanding and evaluating remote sensing retrievals of CER, using probes as an unambiguous truth for CER retrieval validation is challenging.

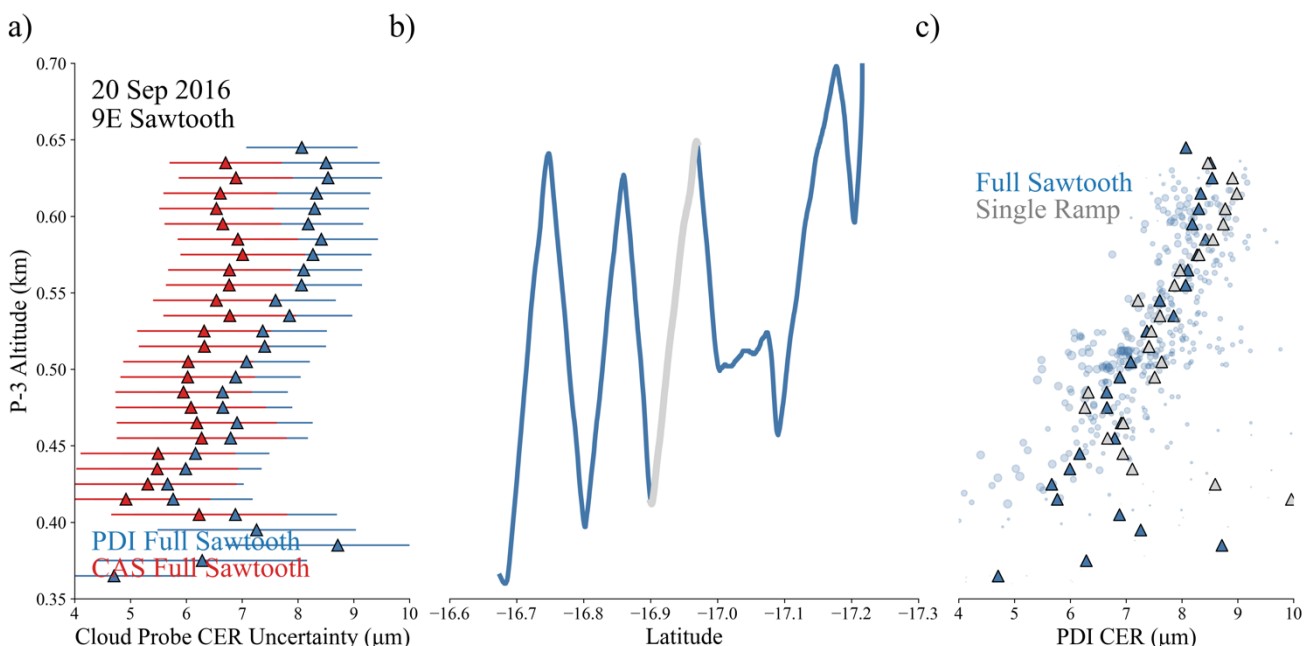


**Figure 22: Cloud probe CER uncertainties and potential sampling biases using the 9E Sawtooth case in Section 4.2 (see Figs. 9-11). (a) Uncertainties in the 10 m layer CER derived from the PDI (blue) and CAS/2D-S (red) DSDs, computed by propagating a ±0.5 µm droplet sizing uncertainty. (b) P-3 flight track (altitude) for the 9E Sawtooth, where a single "ramp" segment is highlighted in**





**gray. (c) CER derived from PDI DSDs when using the single ramp segment (gray) versus using the entire Sawtooth maneuver (blue), highlighting potential biases resulting from how the cloud is sampled.**

The results of this study, and the open questions discussed here, serve as a reminder that cloud microphysics remains a difficult observational problem that additional information content – spectral, polarimetric, in situ, or otherwise – does not always help to clarify. Though spectral and polarimetric observations are complementary, this does not imply that these observations

provide like information on the cloud. They in fact have differing sensitivities and information content, and it is difficult to bridge their respective retrieval spaces using one cloud radiative model. Moreover, the disagreement between the probes used here, and the uncertainties and potential sampling biases involved with their use, imply that an objective microphysical truth in many cases may be elusive.

## 6 Summary

In this paper, we show results of an evaluation of imager spectral retrievals of liquid cloud effective radius (CER) for marine boundary layer clouds from eMAS against co-located polarimetric retrievals from RSP and multiple in situ cloud probes (CAS/2D-S, PDI) obtained during the 2016 deployment of the NASA ORACLES field campaign. A brief overview of eMAS operations during ORACLES 2016 also is included, as are examples of cloud optical property retrieval imagery. In addition to shortwave spectral channels having heritage with MODIS, including the 1.6 and 2.13 µm SWIR channels and 3.7µm MWIR

channel having sensitivity to CER, eMAS has additional SWIR spectral channels having similar, though not identical, CER sensitivity that have not been used in previous campaigns for such retrievals. Moreover, on several occasions eMAS observed the backscatter "glory" region that, having sensitivities similar to the polarized cloud-bow, enables inference of CER and the effective variance (CEV) of the droplet size distribution (DSD) of the cloud; CER and CEV results from one of these cases are shown.


The evaluation is focused on two case studies on 20 September 2016 featuring coordination between the ER-2 remote sensing and P-3 in situ aircraft, where the P-3 was sampling within the cloud while the ER-2 flew overhead. For the first case, a coordination at 9E longitude featuring a "sawtooth" sampling strategy by the P-3 (9E Sawtooth), the eMAS spectral retrievals agree to within roughly 1 µm with the RSP cloud-bow retrievals. For the second case, a coordination at 10.5E longitude

featuring a ramp maneuver through the cloud by the P-3 (10.5E Ramp), there is strong disagreement amongst the retrievals, with the spectral retrievals being 1-3 µm larger than the RSP cloud-bow retrievals. In both cases, the in situ probes also disagreed on CER by over 1 µm, with PDI observing larger CER than CAS/2D-S. Alternate spectral retrieval assumptions were explored for both cases, namely an updated complex index of refraction dataset for liquid water in the SWIR and assumptions on the DSD CEV that are closer to the CEV retrieved by RSP polarimetry. For the 9E Sawtooth case where good

initial agreement was seen, the alternate retrieval assumptions, particularly on CEV, negatively affected retrieval agreement



(see Fig. 11). On the other hand, for the 10.5E Ramp case where the spectral and polarimetric retrievals strongly disagreed initially, the spectral retrievals are brought into closer agreement with RSP polarimetry and PDI using the combination of the updated CEV and refractive index assumptions.

The glory analysis focused on a single case obtained on 14 September 2016. For this case, the eMAS spectral CER retrievals generally were larger than that retrieved from RSP polarimetry. However, matching forward modelled SWIR and MWIR reflectance across the glory, computed for various combinations of CER and CEV, to the observed reflectance implied smaller CER, closer to the polarimetric retrievals, and CEV also closer to the polarimetric retrievals (and smaller than the heritage CEV = 0.1 assumption). Using CEV roughly consistent with these retrieved values yields smaller spectral CER and better

agreement with the polarimetry.

**Code and Data Availability**

The analysis code and supporting datasets used in this paper that are not publicly archived can be obtained from the first author upon request. The standard eMAS Level-1B geolocation/calibrated radiance products are publicly archived at the Level-1 and Atmosphere Archive and Distribution System (LAADS) Distributed Active Archive Center (DAAC) hosted at NASA's

Goddard Space Flight Center (GSFC) (https://ladsweb.modaps.eosdis.nasa.gov/missions-and-measurements/mas/). The RSP polarimetric and spectral cloud products from the NASA Goddard Institute for Space Studies (GISS) team can be found at https://data.giss.nasa.gov/pub/rsp/data/ORACLES-2016/L2CLD/. NASA ORACLES 2016 P-3 data, including the merged dataset products and the merged microphysics products used in this paper, can be found at https://espo.nasa.gov/ORACLES/archive/browse/oracles/id8/P3; the ER-2 data, including the platform housekeeping files

used in this paper, can be found at https://espo.nasa.gov/ORACLES/archive/browse/oracles/id8/ER2.

**Author Contributions**

KM designed and performed the scientific analysis and led the preparation of the manuscript with contributions from all authors. SP and DM provided extensive guidance and insight on the scientific analysis. GTA processed all eMAS cloud retrievals along with the GSFC RSP retrievals. NA provided the RT calculations and other supporting analysis. JS-G and MW

provided guidance on the use and interpretation of PDI data. BC provided guidance on the use and interpretation of RSP data. SG, GM, and JOB provided guidance on the use and interpretation of CAS/2D-S data and the ORACLES microphysics datasets.



**Competing Interests**

The authors declare that they have no conflict of interest.

**Acknowledgements**

The NASA ORACLES mission, including participation and science analysis by all authors except DM and MW, was funded by the NASA Earth Venture Suborbital-2 grant NNH13ZDA001N-EVS2. Additional funding supporting the GSFC team's RSP cloud retrieval activities was directly provided by the NASA Radiation Sciences program. The authors also are extremely grateful for the technical support for eMAS before, during, and after ORACLES 2016 provided by the Airborne Sensor Facility

(ASF) at NASA Ames Research Center (ARC), in particular contributions by R. Dominguez, T. Hildum, P. Grant, K. Dunwoody, J. Myers, T. Ellis, and J. Jacobson.

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
