# Peer review of "Evaluating spectral cloud effective radius retrievals from the Enhanced MODIS Airborne Simulator (eMAS) during ORACLES"

_EGUsphere, 2024_

## Referee Comment (RC2)

**Review of 'Evaluating spectral cloud effective radius retrievals from the Enhanced MODIS Airborne Simulator (eMAS) during ORACLES' by K. Meyer et al.**

In this study, a wide range of cloud droplet effective radius observations are compared, both remotely sensed and in-situ measured by different techniques. Sensitivities to retrieval assumptions are quantified, and possible reasons for agreement or disagreement between the different observations are given. This is an extensive analysis, which has been carried out very carefully, and presented extremely well. It is of great value for the scientific community, not only for those doing (satellite) retrievals or in-situ measurements but also for users of these observations for e.g. cloud-aerosol interaction studies. I have only one major concern, related to the effects of above-cloud absorbing aerosols on the results (see below), which needs to be addressed by the authors. Otherwise, I only have a few minor comments.

**General**

My main comment is about the potential presence of absorbing aerosol (smoke) above the clouds. As mentioned on page 3, 'ORACLES targeted the unique aerosol and cloud environment over the southeast (SE) Atlantic Ocean where an extensive biomass burning smoke layer overlies a quasi-permanent marine stratocumulus cloud deck'. Indeed, on the main day of study, 20 September 2016, extensive smoke appears to have been present in the study region, with absorbing aerosol index (AAI) values (much) higher than 2 (see image below taken from https://www.temis.nl).

[Figure]

Surprisingly, no analysis of this aerosol layer is included in the paper, and it is not taken into account in any of the retrievals. Effects on COT and CER retrievals are discussed on page 17, and it is stated that 'CER retrievals, on the other hand, are substantially less biased, e.g., less

than 5% on a monthly mean scale (Meyer et al., 2015), since the above-cloud aerosol spectral absorption is at a minimum in the SWIR and MWIR (de Graaf et al., 2012; Haywood et al., 2004). However, while it is true that absorption by smoke is minimal in the SWIR and MWIR, CER retrievals are affected through the coupling with COT. Haywood et al. (2004) find CER underestimates of up to 2 $\mu$m at 3.7 $\mu$m, 1 $\mu$m at 2.13 $\mu$m, and as much as 5 $\mu$m at 1.63 $\mu$m. In the context of this paper, these are significant biases, which must be taken into account. The authors either need to include these aerosols in the retrievals or – alternatively – demonstrate that no significant aerosol was present above the clouds in the cases studied.

**Specific**

Fig. 2: The Aqua MODIS comparisons appear to suggest a much larger eMAS degradation than the RSP comparisons. Can you comment on that? Also, a symbol appears to be missing for the Aqua MODIS comparison of the 2.13 micron channel on 20 September.

Fig. 4: I am somewhat surprised by the large difference in retrieval uncertainties between the spectral channels. In particular, the 3.7 $\mu$m CER has a relatively very low uncertainty. Could it be that uncertainties related to estimating the thermal emission contribution to the observed radiance as well as the error of 5% in the solar component (compared to 7% error in the reflectance for the other channels) are judged too optimistically? A related question is that, if I am not mistaken, these uncertainties are not included in the further analysis. For example, in Fig. 11 the whiskers denote spatial variability but single-pixel retrieval uncertainty is not accounted for. If the retrieval uncertainty of the 1.6 $\mu$m CER is really as large as 50% (which seems to be the case in Fig. 4) – corresponding to about 4 $\mu$m and likely a combination of systematic and random errors – this puts the results in Fig. 11 in a different perspective. Can the authors comment on this?

P29, 647-648: I am not sure what this statement means. The inter-wavelength differences seem to be comparable between the two cases:

- Sawtooth: PDI 7.9-8.3 um, CAS 6.5-6.7 um
- Ramp: PDI 9.0-9.4 um, CAS 7.7-7.9 um

P43, Fig. 21: I do not see how CER at 1.62 micron can increase as a result of doubling above-cloud water vapor. Can you explain?

**Technical**

P7, L204: remote -> remotely

P19: There is some duplication between text and caption. In general, content of the figure is best described in the caption. Suggest to transfer some of the description to the caption (e.g., arrows, labels, blue boxes).

P30, L677: spectral -> spectrally

P35, L785-787: It would be good to include the meaning of the vertical dotted lines also (or only) in the caption of Fig. 19.

P42, L909: Prefer: 'it is'.

P43, Fig. 21: Please indicate for clarity what the dotted boxes refer to. I guess these are the 'default' retrieval results from Figs. 11a and 15a?

P44, L954: 'bias difference' sounds strange and 'double'. Consider to replace with 'difference'.

P47, L1051: Incomplete reference

P48, L1087: n/a-n/a. Correct and include doi,

P49, L1103: n/a-n/a. Correct and include doi,

P51, L1185: ldots. Correct and include doi.

P52, L1211: Incomplete reference

P53, L1248: Incomplete reference

P53, L1260: Incomplete reference

P54, L1290: n/a-n/a. Correct and include doi.

---

## Author Comment (AC1)

We thank the reviewer for the thoughtful review and kind words. Our responses to both the general and minor comments, which we hope adequately address the reviewer's concerns, are below.

General comments:

*P11L324 – P11L330: The technique that utilized instrument-A (here RSP) to cross calibrate instrument-B (here eMAS) is very valuable to the community. I suggest [NOT MANDOTORY] adding an appendix to describe the technique in detail (for example, how to deal with different sensor geometries that might involve different attitude correction [uncertainty might become larger than radiometric uncertainty], instrument line shapes etc.) to make the technique 1) citable and 2) generally applicable to cross-calibration for other instruments.*

> This is a good suggestion, though we note that the RSP cross-calibration provides only the temporal degradation (darkening) trend in the eMAS radiometric calibration and is not used as an absolute radiometric benchmark. Instead, the absolute radiometric benchmark was co-located observations from MODIS obtained during targeted under-flights of Aqua with the ER-2 flying along the nadir Aqua ground track. Those eMAS-MODIS comparisons were limited to the near-nadir observations from both imagers for which the observation angles were in rough agreement, and specifically utilized comparisons of the optical property retrievals that inherently account for sensor-specific characteristics (e.g., spectral response functions). The RSP cross-calibration used reflectance comparisons that did not account for spectral response differences and thus cannot provide an absolute radiometric benchmark. We've made some text additions to section 3.2 that we hope provide more clarity. We also are in the process of drafting a calibration readme document that will be posted in the LAADS data archive along with the L1B, as has been done for previous campaigns.

*Based on the AMT reference guideline, If the author's name is part of the sentence structure only the year is put in parentheses ("As we can see in the work of Smith (2009) the precipitation has increased") E.g., P8L242 (Platnick, 2000) to Platnick (2000), P8L248 (Gupta et al., 2022b), P9L269 (Gupta et al., 2022b) etc. Additionally, please make the reference order consistent throughout the paper (e.g., when multiple references are used, oldest reference goes first).*

> Good catch. We relied on the Copernicus style template in Zotero for our references and citations, which greatly eased reference management but that obviously does not completely adhere to the AMT style guide. We've modified what we think are all the problematic citations, but can't guarantee that we didn't miss a few.

*From measurement perspective, accounting for the vertical heterogeneity (z direction) of clouds while constraining the horizontal heterogeneity (xy direction) is quite challenging (which might explain the different results between sawtooth and ramp cases). I wonder what the author think is the better approach? Will the square spiral into the clouds help the constraint of horizontal heterogeneity of clouds?*

> This is an excellent question. As with any in situ sampling approach, trade-offs abound. The "sawtooth" (or porpoise) sampling shown in our first case study maximizes the vertical sampling statistics, though the horizontal sampling is limited to the direction of flight and, depending on the heterogeneity of the cloud, may not be adequate. A square spiral may improve horizontal sampling, though the sampling of different vertical layers likely will be spatially decoupled. Ultimately, the sampling approach should be dictated by the science objectives. For evaluating remote sensing retrievals, particularly from sensors having coarser spatial resolution (e.g., spaceborne imagers, etc.), the question to be answered is can a single aircraft provide a representative sample of the retrieval domain, which is complicated by spatial and temporal changes in the cloud. We think this remains an open question.

Minor comments:

*P5L134: What does "effective pixel size" mean? Field of view at different angles?*

Correct, "effective pixel size" acknowledges the differences in the projected size of the instantaneous field of view that is a function of viewing angle and the altitude at which co-location of the angular observations occurs (e.g., ground, cloud-top).

*P5L154: change to "have heritage with the cloud products … from MODIS science team" or something similar*

Done.

*P8L226: "sun/satellite" to "sun/sensor"*

Done.

*P8L227: Since at P41L886 the above-cloud gaseous absorption corrections were discussed, it would be good if some clarifications on the definition of "top-of-cloud reflectance" (whether gases and/or aerosols are taken into account) here.*

Good suggestion. We've added the definition to P8L238-239, and also have added a reference to "top-of-cloud reflectance" to the atmospheric correction discussion on P42L999-1000.

*P11L308: change "with some degree of confidence" to "(with some degree of confidence)"*

Done, though using commas instead of parentheses (P11L319).

*P15L385: In Figure 4, rightmost panel for CER (3.7 mm), a very different pattern (bright potion, relatively low retrieval uncertainty) shown up on the upper side, what can be the causes?*

Good observation! That bright pattern corresponds to the region of smaller effective radius retrievals in all three spectral channels apparent in the corresponding right three panels of Figure 3. Since the retrieval uncertainties essentially are proportional to the sensitivity of the solution space to perturbations in the observed top-of-cloud reflectance (with the size of the perturbations determined by the magnitude of the impacts of the various error sources), the 3.7μm solution space is less sensitive to reflectance errors (perturbations) at these small sizes than it is at larger sizes, and this difference in sensitivity across sizes is much larger than it is in either the 2.13 or 1.62μm solution spaces.

*P21L475: It looks like the CAS$_{shifted}$ was moving into the right direction to agree with PDI. Is it possible to use a different constraint to shift size bins so CAS matches PDI and claim that the in situ can achieve agreement? Or the constraint will become nonphysical?*

CAS$_{shifted}$ does move in the right direction with respect to PDI, but it's important to remember that PDI also has inherent uncertainties/biases such that neither probe can be considered an unambiguous truth, as we state in the paper. Different probes see different collections of drops in the same cloud due to numerous factors, many of which depend on measurement principle (sizing and counting uncertainty) and probe design (drop size range sampled, size of sample volume, etc.). Thus it's extremely difficult to "correct" for these differences to achieve agreement between the probes, since as the reviewer suggests, agreement may be achieved for one moment of the DSD but unphysical results for another.

*P33L751: "misleadingly intensifying the appearance of scene heterogeneity": I think this statement is valid for visible. The appearance of heterogeneity should be solid as they come from the NIR or SWIR channels. In return, the cloud optical thickness (visible) will not contain large 3D bias, but CER (NIR) might still suffer from the 3D effects.*

Agreed, there is indeed heterogeneity in the SWIR channels for this scene that influence the CER retrievals. The statement in question here was made simply to point out that we stretched the color scale for each spectral channel in this false color RGB such that we enhance the visibility of the glory, yet at the expense of enhancing also the contrast between relatively darker and brighter pixels outside the glory – to the extent that the "darker" cloud pixels look like cloud-free regions rather than just less bright clouds.

*P36L807: "10mm" to "10 mm" (adding a space in-between)*

Good catch! This is fixed.

---

## Author Comment (AC2)

We thank the reviewer for the thoughtful comments and hope that our responses adequately address the concerns, particularly regarding the impacts of absorbing aerosols on the bi-spectral CER retrievals via solution space non-orthogonality. Our responses to the general, specific, and technical comments are below.

General comments:

*My main comment is about the potential presence of absorbing aerosol (smoke) above the clouds. As mentioned on page 3, 'ORACLES targeted the unique aerosol and cloud environment over the southeast (SE) Atlantic Ocean where an extensive biomass burning smoke layer overlies a quasi-permanent marine stratocumulus cloud deck. Indeed, on the main day of study, 20 September 2016, extensive smoke appears to have been present in the study region, with absorbing aerosol index (AAI) values (much) higher than 2 (see image below taken from* [https://www.temis.nl](https://www.temis.nl)*).*
*Surprisingly, no analysis of this aerosol layer is included in the paper, and it is not taken into account in any of the retrievals. Effects on COT and CER retrievals are discussed on page 17, and it is stated that 'CER retrievals, on the other hand, are substantially less biased, e.g., less than 5% on a monthly mean scale (Meyer et al., 2015), since the above-cloud aerosol spectral absorption is at a minimum in the SWIR and MWIR (de Graaf et al., 2012; Haywood et al., 2004). However, while it is true that absorption by smoke is minimal in the SWIR and MWIR, CER retrievals are affected through the coupling with COT. Haywood et al. (2004) find CER underestimates of up to 2 µm at 3.7 µm, 1 µm at 2.13 µm, and as much as 5 µm at 1.63 µm. In the context of this paper, these are significant biases, which must be taken into account. The authors either need to include these aerosols in the retrievals or – alternatively – demonstrate that no significant aerosol was present above the clouds in the cases studied.*

The reviewer raises an important point on the non-orthogonality of the COT/CER solution space for these bi-spectral retrievals, particularly for those using 1.6x µm versus 2.x µm or 3.7 µm. As the reviewer correctly states, above-cloud aerosol absorption that attenuates only the VIS/NIR channels nevertheless can cause changes to CER retrievals when the reflectance observations lie in portions of the COT/CER solution space that are non-orthogonal. Non-orthogonality of the COT/CER solution space typically is dependent on the SWIR/MWIR wavelength (1.6x µm is the most non-orthogonal; 3.7µm is least) and on the COT of the cloud (non-orthogonality increases with decreasing COT). While we did consider the impacts of above-cloud absorbing aerosols using regional/seasonal retrieval statistics from previous studies focusing on the SE Atlantic biomass burning smoke, we quite frankly are embarrassed that we did not consider their impacts in the specific case studies shown here.

We now have included estimates of these impacts for the 9E Sawtooth and 10.5E Ramp probe comparison case studies in Section 4.2, and specifically only for the 1.6x and 2.x µm retrievals. While the reviewer correctly notes that Haywood et al. (2004) finds CER impacts up to 2 µm for the 3.7 µm channel (see Haywood's Fig. 3(b) and associated text showing impacts of above-cloud smoke), these strong impacts for the NIR/3.7 µm channel pair are evident only for larger CER, roughly apparent (i.e., retrieved) CER > 16 µm (note: the same cannot be said for the above-cloud dust case in Haywood's Fig. 4, where the 3.7 µm reflectance also changes, thus 3.7 µm CER retrievals show impacts throughout the solution space). In our two case studies, apparent CER is < 10 µm for the Sawtooth case and < 12 µm for the Ramp case, both squarely in the orthogonal portion of the solution space and, using Haywood's Fig. 3 (b) as a guide, likely to have little change when accounting for above-cloud aerosol absorption in the NIR 0.87 µm channel used for the eMAS retrievals in our study.

Specific details on this additional analysis have been added to Section 4.2 (and updates to Figs. 11 and 15), along with smaller text additions at the end of Section 4.1, in the Section 5 discussion, and in the Section 6 summary. We also have replaced Figure 12 with a new figure showing the COT/CER solution spaces for a cloud only and a cloud with overlying aerosol having AOT defined by co-located HSRL-2 retrievals. The upshot is that for the 9E Sawtooth case, we estimate that the CER retrievals can increase by roughly 3.3 and 4.1 µm for the 1.62 and 1.66 µm channels, respectively, and by roughly 1.0 µm for all 2.x µm channels, when accounting for the above-cloud aerosol absorption in the NIR only. These findings are consistent with Haywood et al. (2004). For the 10.5E Ramp case, however, the 1.62 and 1.66 µm retrievals increase only by 1.1 and 1.5 µm, respectively, while the impacts on the 2.x µm retrievals are less than 0.2 µm, or essentially negligible. In both cases, the above-cloud AOT from HSRL-2 is near 0.6, which cannot explain the differences in the aerosol impacts between the two cases. Rather, the 10.5E Ramp case is optically

thicker (COT ~23.3) such that the observed spectral reflectance is in the more orthogonal portion of the solution space (thus smaller impacts), while the 9E Sawtooth case is optically thinner (COT ~9.4) such that the observations are in the less orthogonal portion of the solution space.

We sincerely thank the reviewer for this insightful comment on an important omission from our study!

Specific comments:

*Fig. 2: The Aqua MODIS comparisons appear to suggest a much larger eMAS degradation than the RSP comparisons. Can you comment on that? Also, a symbol appears to be missing for the Aqua MODIS comparison of the 2.13 micron channel on 20 September.*

The reviewer has a good eye! The MODIS comparisons do appear to suggest a larger degradation, though we note that these comparisons are done only for the small geographic region where the ER-2 was spatially/temporally co-located with the Aqua nadir ground track, thus are a small sample size that may have larger day-to-day variability. Nevertheless, we historically have chosen MODIS as our radiometric benchmark given our desire for consistency between the eMAS retrievals and their spaceborne MODIS counterpart, and the comparison itself involves iterating the eMAS calibration until COT/CER retrieval consistency is achieved for the co-located near-nadir observations. The comparison against RSP, on the other hand, involves reflectance comparisons that do not account for spectral response differences (thus cannot be used as absolute radiometric benchmarks), but that do use data sampled from each entire science flight that inherently includes a wider variety of scenes and geographic sampling and that can provide a relative degradation curve covering the entire campaign. We further note that, while an eMAS-RSP retrieval comparison approach similar to the eMAS-MODIS approach is possible, it may yield calibration adjustments that do not enable COT/CER retrieval consistency with MODIS.

Regarding the 20 September 2.13µm comparison, the CER retrievals used for the iterative calibration adjustment approach had some outliers that muddied the results. Therefore we excluded this day for this channel only, which explains the missing symbol in Fig. 2. That said, we expect similar behavior in this channel as in the 1.62µm channel (i.e., no degradation between 18 and 20 September) given that both are on the same focal plane; the VNIR channels that do show day-to-day degradation are on a separate focal plane.

*Fig. 4: I am somewhat surprised by the large difference in retrieval uncertainties between the spectral channels. In particular, the 3.7 µm CER has a relatively very low uncertainty. Could it be that uncertainties related to estimating the thermal emission contribution to the observed radiance as well as the error of 5% in the solar component (compared to 7% error in the reflectance for the other channels) are judged too optimistically? A related question is that, if I am not mistaken, these uncertainties are not included in the further analysis. For example, in Fig. 11 the whiskers denote spatial variability but single-pixel retrieval uncertainty is not accounted for. If the retrieval uncertainty of the 1.6 µm CER is really as large as 50% (which seems to be the case in Fig. 4) – corresponding to about 4 µm and likely a combination of systematic and random errors – this puts the results in Fig. 11 in a different perspective. Can the authors comment on this?*

This is a great observation. Since we're using the same code base as the MODIS MOD06 and MODIS/VIIRS CLDPROP optical property retrievals, the uncertainty handling in the eMAS retrievals shown here is identical to those satellite products and, for the 3.7 µm channel, does include thermal emission removal uncertainty components, specifically the solar irradiance and surface/cloud temperatures (see Section III/F in Platnick et al. (2017)). The 3.7 µm uncertainties are smaller largely due to the mostly orthogonal solution space, in particular for the COT/CER ranges observed in this scene. We note that similar differences between the 3.7 and 2.1µm CER retrieval uncertainties are found in the MOD06 uncertainties (see Fig. 14, Platnick et al. (2017)), though perhaps are less extreme since the MODIS radiometric uncertainty has less spectral variation than the 5% vs 7% uncertainties used here for eMAS. The rationale for the different SW vs MWIR radiometric uncertainty assumptions is an attempt to account for the different calibration sources for each spectrum; the SW, as described in our answer above and in the text, uses lab sphere calibrations and comparisons with other imagers, while the MWIR, like the thermal

IR, is calibrated using an onboard blackbody that is expected to have greater stability. The reviewer is indeed correct that the pixel-level uncertainties are not included in the further analysis.

Regarding the 1.6x µm uncertainties, the reviewer makes an important point. Correctly understood, these retrieval uncertainties essentially are sensitivities to changes in spectral reflectance such that, all error sources being equal across retrievals, more non-orthogonal solution spaces will have higher retrieval uncertainties. As such, all the results in this paper should be interpreted in light of these uncertainties. Thus, along with the text additions describing the above-cloud aerosol impacts, we have added statements on the higher uncertainties associated with the 1.6x µm retrievals that, like the impacts of the above-cloud aerosol absorption in the NIR, are directly related to its less orthogonal solution space (e.g., P25, L613-616).

*P29, 647-648: I am not sure what this statement means. The inter-wavelength differences seem to be comparable between the two cases:*
  *- Sawtooth: PDI 7.9-8.3 um, CAS 6.5-6.7 um*
  *- Ramp: PDI 9.0-9.4 um, CAS 7.7-7.9 um*

Good catch. This statement on the PDI adiabatic signature differences between the two case studies referred to an older iteration of the data/table that we later found to be in error. We have removed the statement from the text.

*P43, Fig. 21: I do not see how CER at 1.62 micron can increase as a result of doubling above-cloud water vapor. Can you explain?*

This result does indeed differ from what one might expect, and we will admit it also caused us to scratch our heads upon first seeing it. However, the explanation lies in spectral water vapor absorption differences coupled with the non-orthogonality of the solution space for the VNIR/1.62 µm retrievals. First, water vapor is only weakly absorbing in the 1.62 µm channel and is stronger in the other SWIR channels and the 0.87 µm channel used for COT retrievals. For the NIR/1.62 µm channel pair only, the atmospheric correction acts to increase the 0.87 µm reflectance, with only a small increase in the 1.62 µm reflectance, thus moving the observation point more "right" than "upwards/rightwards" in the solution space. Using the LUT plot in Fig. 12 (a) as a guide, this rightward movement paradoxically yields larger CER retrievals. For all other retrieval channel pairs, the atmospheric correction has larger impacts on the SWIR channels, thus yielding larger SWIR reflectance and smaller CER.

Technical comments:

*P7, L204: remote -> remotely*

Done.

*P19: There is some duplication between text and caption. In general, content of the figure is best described in the caption. Suggest to transfer some of the description to the caption (e.g., arrows, labels, blue boxes).*

Done.

*P30, L677: spectral -> spectrally*

Spectral in this case refers to the weighted CEV for each spectral channel; the CEV itself is vertically weighted. We have changed the text to clarify this.

*P35, L785-787: It would be good to include the meaning of the vertical dotted lines also (or only) in the caption of Fig. 19.*

We have added the explanatory text to the Fig. 19 caption.

*P42, L909: Prefer: 'it is'.*

Changed.

P43, Fig. 21: Please indicate for clarity what the dotted boxes refer to. I guess these are the 'default' retrieval results from Figs. 11a and 15a?

> The reviewer is correct, these correspond to the "default" retrieval results in Figs. 11(a) and 15(a). We have added explanatory text to the figure caption.

*P44, L954: 'bias difference' sounds strange and 'double'. Consider to replace with 'difference'.*

> Changed.

*P47, L1051: Incomplete reference*

> This reference refers to an online document at the Ames ASF website for eMAS, whose url apparently does not get included when using the Zotero bibliography template for Copernicus publications. We will add this by hand.

*P48, L1087: n/a-n/a. Correct and include doi*

> Fixed.

*P49, L1103: n/a-n/a. Correct and include doi.*

> Fixed.

*P51, L1185: ldots. Correct and include doi.*

> Fixed.

*P52, L1211: Incomplete reference*

> See comment above on our use of the Copernicus template with Zotero. We will add the url by hand.

*P53, L1248: Incomplete reference*

> We will add the url by hand.

*P53, L1260: Incomplete reference*

> Fixed.

*P54, L1290: n/a-n/a. Correct and include doi.*

> Fixed.

---

## Author Comment (AC3)

We thank the editor for her interest in our paper and hope that our responses to her comments adequately address the concerns raised. We also note that her concerns regarding the above-cloud absorbing aerosol impacts on CER retrievals were also raised by Reviewer 2. We address those concerns in our response to that reviewer, and our responses here often refer to those.

General comments:

*The introduction would benefit from a stronger tie-in to the cloud-aerosol interactions motivating ORACLES, currently it is highly general. Specific motivations include smoke-cloud microphysical interactions; the time evolution in CER towards discriminating the boundary-layer semi-direct effect from the microphysical interactions; the impact of intervening absorbing aerosol on the CER retrievals.*

> We appreciate the editor's suggestion. That said, this paper has a narrow focus on the microphysical retrievals, and probe observations, that have relevance beyond ORACLES and the specific motivations for that campaign. As such, we'd prefer to keep the introduction tailored to that narrow focus and let the Redemann et al citation do the lifting on the details/motivations of ORACLES itself.

*Overall I was surprised to not see more language about overlying smoke in the manuscript and its cases. Did I miss something? Was this not an issue?*

> We thank the editor for raising this concern, which Reviewer 2 did also. While we did consider above-cloud aerosol effects on a regional/seasonal statistical basis, we regret that we did not consider these effects on the specific case studies shown here. Please see our response to Reviewer 2 for a detailed discussion of the impacts of above-cloud aerosols on the CER retrievals, an additional analysis we performed to account for these impacts in the case studies shown, and subsequent changes made to the text and figures.

Specific comments:

*Paragraph spanning lines 56 to 69: why aren't SSA assumptions of the overlying smoke on the CER retrievals mentioned here?*

> The discussion in this paragraph primarily is focused on cloud radiative model assumptions themselves. That said, in response to the Reviewer 2, we have added text on forward model errors due to ignoring above-cloud absorbing aerosol layers to Sections 4.1, 4.2, 5, and 6 that we think addresses the concern you raise here.

*In the case of ORACLES the overlying smoke layer was also humid enough to support clouds at times near the top of the aerosol layer. The RSP also provided useful retrievals of these mid-level cloud properties, as shown in Adebiyi et al 2020. This paper would be worth citing, perhaps at the end of line 172 or as an example of CER retrievals helping to determine aerosol-cloud interactions in the ORACLES regime.*

> We thank the editor for reminding us of this paper and have taken her suggestion to add the citation to the end of line 172 (now line 174), along with some minor text modifications.

*Line 141: what is the lower limit of the drop size detected by CAS? Is it also 3 micron? It is also my understanding that the smallest dropsize bin of the PDI was too noisy to use, perhaps because of interference from the airplane's electrical system. Can the ORACLES PDI PI comment? For the thin, polluted clouds sampled during ORACLES, the lower size limits will be important. How did the Nds compare?*

> While CAS can detect particles as small as 0.5μm diameter, for the ORACLES Microphysics dataset a lower limit of 3μm diameter was applied to avoid counting aerosols as cloud droplets. We note also that, for the merged DSDs, a lower limit of 50μm diameter is applied to 2D-S. More information on the probe size ranges/limits, as well as campaign-wide comparisons of probe Nd, LWC, etc., can be found in the O'Brien et al. readme file cited in the paper.

Regarding potential noise interference from the aircraft's electrical system on the smaller size bins of PDI, this was only an issue in the 2017 and 2018 deployments; PDI did not suffer this issue in the 2016 deployment data used in this study.

*Section 2.4: continue to be puzzled why the SSA of the overlying smoke layer isn't mentioned as an additional uncertainty within the eMAS retrieval.*

See our response to Reviewer 2 and to your comments above.

*Top of p. 10: eMAS issues seem like they belong in data section.*

We respectfully disagree, as this section is intended to highlight specifics of the eMAS operations during the ORACLES 2016 deployment, whereas the data section is intended to provide general details on the retrieval products and their heritage/theoretical basis.

*Fig 1 caption: provide total # of flights in each category.*

Good suggestion. Done.

*Line 419: in combination, for LWP, the bias is about 10% (Grosvenor et al. 2018).*

Indeed, biases in COT and CER will propagate to LWP calculations that, for bi-spectral imager retrievals, are proportional to the product of COT and CER.

*Figs 7-9: What was the AOD overlying the cloud in both 20 Sept cases? Same question applies to the 14 Sept case later on.*

The AOD overlying the cloud in both 20 September case studies, obtained from co-located HSRL-2 curtains, was just under 0.6, which we now indicate in the text; for the 14 September flight, HSRL-2 did not operate. As we discuss in our text additions on the impacts of this above-cloud aerosol layer, the specific impacts on the CER retrievals due to NIR absorption and solution space non-orthogonality act to increase CER, though to varying degree.

*Fig. 10: are the cloud DSDs unimodal throughout the vertical column? Might be good to examine the actual DSDs near cloud top to understand the differences better. Maybe the issue is that the 2DS is having trouble picking up drops near the smaller end of its range, so that the DSD looks more bimodal within the CAS/2DS combined distribution. Is there any evidence of coincidence undercounting by the CAS? It would also be good to show the vertical profiles in LWC, allowing you to add in the bulk King-derived LWC. How thermodynamically well-mixed was the cloud layer?*

While we share the editor's interest in understanding why the cloud probes differ, this paper is focused primarily on the remote sensing retrievals of CER and the impacts of bi-spectral retrieval assumptions on CER agreement (or disagreement). As such, we think that attempting to diagnose differences in the probe datasets is well beyond the scope of this paper. We simply take the probe data as they are provided in the public ORACLES archive. We do note, though, that analyses that include the campaign-wide LWC profiles from the probes in question are shown in the O'Brien et al document, cited in the paper, describing the ORACLES Microphysics datasets. Furthermore, we do agree that this subject perhaps deserves a dedicated study, and have added a statement to this effect to the discussion in Section 5 (lines 1058-1060).

*Table 2: why are the polarimetric optical depths so low?*

These are vertically weighted COTs, or essentially the optical depth within the cloud where each retrieval has its peak sensitivity. Since the polarization signal is a single-scattering phenomenon, its sensitivity lies at the very top of the cloud, as shown by its corresponding weighting function in Fig. 10 (c). The spectral retrievals, on the other hand, have peak sensitivity much deeper in the cloud, a fact that drives the discussion on the impacts of vertical heterogeneity.

*p. 41: how confident are you in the CO2, CH4, H2O above-cloud optical depths/loadings? If I understand correctly these are coming from a reanalysis - is that MERRA2? You mention doubling the concentrations, these might bring you closer to the actual values. How do the reanalysis H20 values compare to those measured on the aircraft? This error source should be genuinely quantifiable using the ORACLES H2O in-situ dataset in contrast to the statement made on line 909. THis may be addressed in Pistone et al 2024.*

>This is an excellent question. The short answer is that we are not confident in these column loadings. For the eMAS retrievals (and their spaceborne MODIS and VIIRS counterparts), we use NCEP GDAS reanalysis profiles for $H_2O$, and standard atmospheric profiles for the remaining gases. This is an obvious source of error in our atmospheric corrections that was the rationale for investigating the sensitivity to these assumptions and their impacts on the CER retrievals. That said, rather than attempting to estimate the exact biases in these reanalysis column loadings, we instead chose to estimate the impacts of extreme biases, here 100% errors that we assume are larger than the actual biases. Even under these extreme error assumptions, the impacts of atmospheric correction errors cannot explain the retrieval differences found in the case studies.

p. 44: more effort can and should be made to understand the differences between the two probes. How about picking a long level incloud-leg, ideally near cloud top, and comparing an average of the full DSD. Is there evidence of bimodality? How does Nd compare? What is the flow volume of the 2 probes? Is the sensitivity to the smallest drop sizes the same?

>Again, while we share the editor's interest in understanding the differences between the probes, we think attempting to answer these difficult questions here is well beyond the scope of this paper. We do agree, though, that such an investigation is of interest to the community, particularly the users of the ORACLES datasets, and should be pursued separately, and have added a statement to this effect in Section 5.

*Section 5: why isn't the aerosol overlying the cloud mentioned for these cases? Could it be neglected? Did you select 2 cases with no overlying AOD? I don't recall reading that and wonder if I missed it.*

>See our responses above and to Reviewer 2 that detail additional analyses and text that we think now address this concern.

*Section 6: Stronger guidance should be provided here to help potential users of the publicly-available eMAS cloud property retrievals. Which one would you recommend?*

>This is a good suggestion. We have added a paragraph at the end of Section 5 that provides advice to users.